# From colorblind to systemic racism: Emergence of a rhetorical shift in higher education discourse in response to the murder of George Floyd

Noor Toraif[1‡], Neha Gondal[2‡]*, Pujan Paudel[3], Alison Frisella[4]

1 School of Social Work, Boston University, Boston, Massachusetts, United States of America,
2 Department of Sociology and Faculty of Computing & Data Sciences, Boston University, Boston, Massachusetts, United States of America, 3 Department of Electrical and Computer Engineering, Boston University, Boston, Massachusetts, United States of America, 4 Lesley University, Social Sciences Department, Cambridge, Massachusetts, United States of America

‡ NT and NG are joint first authors on this work.
* gondal@bu.edu

**Data Availability Statement:** All statements and attribute data are available via The University of Michigan's openICPSR data repository at https://

## Abstract

We use topic modeling and exponential random graph models (ERGM) to analyze statements issued by Institutions of Higher Education (IHEs) (N = 356) in the United States in the aftermath of George Floyd's murder in May 2020. Prior research investigating discourse on race in IHEs demonstrates the prevalence of two paradigms. First, the ideology of 'colorblind racism' treats systemic racism—a form of racism where social, political, and economic institutions are organized in a way that disadvantages people of color—as having largely existed in the past. Consistent with this, IHE responses to prior race-related incidents on campus have emphasized individual prejudice, avoiding discussion of systemic racism. Second, 'diversity' orthodoxy, which treats race as a cultural identity and emphasizes the instrumental benefits of racial heterogeneity on campus, is commonplace in IHEs. Topic modeling of statements issued in 2020 reveals the prevalence of several themes including the systemic and enduring nature of racism in the United States, diversity orthodoxy, humanist responses reflecting rhetoric consistent with colorblind racism, and COVID-19 response strategies. ERGM reveals fragmentation in the discourse based on IHE attributes. Religiously affiliated IHEs and those located in Republican-voting states attend more to diversity and humanist discourse, and less to systemic racism. Elite IHEs, those in Democrat-voting states, and IHEs with high percentages of Black students are more focused on systemic racism. Overall, as compared to colorblind racism and diversity orthodoxy established in prior work, our analysis reveals two striking rhetorical shifts on race discourse in IHEs in the aftermath of George Floyd's murder: (1) from a colorblind ideology to discussing the systemic nature of racism in the United States, and (2) from acknowledging perpetrators but not the broader context of racism in on-campus incidents to acknowledging diffuse racism manifest in society but refraining from explicitly naming any wrongdoers.

www.openicpsr.org/openicpsr/project/192941/
version/V1/view.

**Funding:** The author(s) received no specific funding for this work.

## Introduction: Discourse on race in institutions of higher education in the United States

Research investigating contemporary discourse on race in the United States highlights the pervasiveness of two paradigms–the ideology of colorblind racism and diversity-based rhetoric [1–6]. Colorblind racist ideology, Bonilla-Silva [1] has influentially argued, contributes to maintaining systemic racism at a time when individual-level racism is deemed to be socially unacceptable. Sociologists consider racism to be systemic when social institutions such as those in the legal, educational, and political spheres are fundamentally organized in ways that disadvantage people of color. The ideology of colorblind racism (also referred in the literature more simply as colorblind ideology, colorblindness, or colorblind racism [e.g., 2, 5, 6]) relies on several frames–means used by persons for interpreting the role of race in society–that contribute to maintaining such systemic inequalities. First, despite indisputable evidence of ongoing race-based inequalities, liberal values are used to justify individualist meritocracy and oppose policies set to redress those disparities. Second, race and racism are considered to be less salient in the present time and, instead, racism is positioned as largely having existed in the past or in isolated incidents. As such, contemporary incidents of racism are attributed to individual prejudices rather than structural forces and power differentials. Third, race-based inequalities are explained away on account of 'natural' predilections or so-called cultural differences.

In interpersonal interactions, colorblind racism manifests through the denial or minimization of the role of race in shaping an individual's experiences or outcomes, what Sue et al. [7] describe as microinvalidation, a form of microaggression. Microinvalidations involve denying the feelings, perceptions, observations, or realities of people of color, processes that contribute to reproducing colorblind racism by suppressing the effects of racism and making it more difficult to identify [8]. This typically occurs through assertions that any advantages or disadvantages sustained by a social group are obtained through merit or its lack, rather than privileges or disprivileges associated with racial identity [1, 2, 5, 6]. Colorblind ideology has been shown to be commonplace in educational institutions in the United States especially with reference to student experiences and attitudes [e.g., 9–12]. Poteat and Spanierman [11] and Worthington et al. [12], for example, find that racially privileged students on university campuses are more likely to rely on colorblind race frames. More specifically, Bonilla-Silva and Forman [9] show that, rather than being openly racist, as was normative in the pre-civil-rights era, racially privileged college students today are adept at couching their racist views through the use of semantic moves such as expressions of ambivalence and invocation of meritocracy when discussing race. Lee [10] shows that Black students are also not immune from colorblind ideology, often invoking cultural explanations for race-based inequalities.

Much like responses of students, similar themes, reliant on colorblindness, are evident in institutional responses to racist incidents on campus and in society, more broadly [13–20]. In analyzing statements released by leaders of K-12 schools in response to racist incidents involving school students or staff, Bridgeforth [20], for example, notes the use of several colorblind frames including denial of the racialized nature of the event and interpretation of the incident as being attributable to individual biases rather than systemic issues. In the context of responses issued by authorities in the aftermath of instances of racism on college campuses, Cole and Harper [13], likewise, find that few of the issued statements acknowledge the role of systemic racism in the incidents. Instead, most statements tend to focus on the perpetrators of violence, thereby shifting attention away from the racialized nature of the incident. Simply put, "[c]ollege presidents are oftentimes willing to address the racist but rarely the racism" [13 p. 326]. More generally, the research suggests that avoidance of racism as a social problem is

consistent both with Bonilla-Silva's [1] colorblind racism ideology as well as the centering of individualistic issues over societal ones.

In contrast to colorblindness, diversity, the second dominant theme in race discourse in the United States today, underscores the significance of race rather than minimizing it. Generally speaking, 'diversity' has been used by organizations, including IHEs, to refer to heterogeneity of persons based on a myriad of social and personal differences such as race, gender, ethnicity, nationality, and disability status [21–25]. In this context, akin to ethnicity, race is framed as a valued 'cultural identity' and racial differences, much like ethnic ones, are viewed as a matter of *cultural* heterogeneity associated with variability in behaviors, expressions, beliefs, and practices. Racial diversity is thus seen as creating conditions for heterogeneity of interactions among community members, which, in turn, are framed as generative of instrumental benefits such as a superior social climate and creativity of thought [26]. This interpretation of diversity, used to showcase commitment to multiculturalism and appreciation of racial differences, has been shown to be in widespread use, arguably 'enshrined,' across organizations and higher educational institutions in the United States [e.g., 21–25]. Berrey [26] describes the institutionalization and legitimization of rhetoric and policies surrounding this understanding of diversity as a new 'orthodoxy' on university campuses. As distinct from an ideology, which provides a template for the organization of the world [1], an orthodoxy constitutes a set of widely shared ideas, beliefs, and practices that guide institutional discourse as well as policy, strategy, and action.

Berrey [26] argues that, over the last two decades of the twentieth century, "diversity" became a keyword in United States Institutions of Higher Educations' (IHEs) policies and programs surrounding race. This shift occurred, in part, due to organizational pressures in a changing political, demographic, and legal climate. An early impetus can be traced to a minority opinion issued in a significant legal case challenging affirmative action admissions policies in the late seventies. This case laid the groundwork for using diversity as a rationale for race-conscious admissions and subsequent contentious lawsuits, both challenging and supporting such policies, helped codify language surrounding diversity. Thereafter, shifts in demographics of the college-going population–a rise in immigrants, foreign students, women, and people of color–generated greater need for strategy and rhetoric to manage increasingly heterogeneous student populations. These strategies diffused rapidly across IHEs, becoming normative and exerting pressure on others to signal their own commitment to inclusiveness. Indeed, diversity acquired so much popularity over time that it came to replace the formerly reigning buzzword, 'multiculturalism,' in higher education rhetoric [24].

Significantly, much like multiculturalism, diversity discourse in IHEs came to signify not only differences based on racial identities but also other attributes such as gender and ethnicity [26]. Research additionally shows that IHEs draw not only on language surrounding diversity, but also allied terms such as equity, democracy, and inclusion [22, 27, 28]. Iverson [22] elaborates on four distinct diversity discourses employed by IHEs: access, disadvantage, democracy, and marketplace. The access and disadvantage frames position students of color as normative outsiders and perpetually 'at-risk' during their time at school. The democracy frame presents diversity as a democratic value by invoking the language of civic responsibility that encourages students to be involved in producing change. Finally, the marketplace frame posits diversity and people of color as commodities that increase the reputational value of an institution. Urciuoli [28], for example, argues that IHEs use this framing of diversity as part of their brand in marketing materials to signal strength and competitiveness on the job market. Race-based diversity is, thus, framed as offering benefits to all students, not only racially marginalized groups, by improving the overall college-going experience.

Despite fundamental distinctions between colorblind ideology and diversity orthodoxy, research shows that the two forms of discourse can coexist in university settings. Warikoo and de Novais [5], for example, explain that undergraduate students tend to rely on colorblind ideology in their pre-college years, but their experiences at university are instrumental to the development of diversity orthodoxy. The authors also find that a small fraction of students adheres to what they call the 'power analysis' race frame. This frame, facilitated by higher education as an institution engaged in 'critical race' pedagogy and scholarship, invokes an analysis of racial injustice and inequity, focusing on the ways in which political, social, economic, and cultural institutions reproduce racism [29–31]. Two dominant theories of race–Racial Formation Theory (RFT) [4] and Critical Race Theory (CRT) [32]–have been especially influential in the development of this discourse [e.g., 33–35]. The main tenet of RFT, developed by Omi and Winant [4], is that categories of race are socially constructed and, consequently, their contents have varied historically and across social structures. Moreover, RFT holds that categories of race, and the racial discrimination based on these shifting categories, have deep historical roots in the United States with profound consequences for the maintenance and reproduction of social, political, and economic inequalities. Finally, identities and social hierarchies generated from racial classification have been and continue to be sites of political struggle and conflict.

Akin to RFT, CRT also treats race as a socially constructed phenomenon and posits that, rather than being rare or individualistic, racism is systemic and pervasive–a common everyday experience of people of color in the United States [32]. Building on the work of legal scholars such as Derrick Bell, Kimberlé Crenshaw, and others, the second major tenet of CRT, 'interest convergence,' holds that, despite their purported neutrality, institutions and laws serve the interests of dominant races and classes and significant racial progress, including legal gains, has only occurred in the United States when the interests of African Americans have coincided with those of white people [36, 37]. CRT also draws our attention to the intersectionality of identities cutting across attributes such as race, class, and gender, which, in turn shape experiences of marginalization and oppression. Finally, the 'voice-of-color' thesis holds that experiences of discrimination accord marginalized groups with a 'competence' to speak about race and racism, which members of dominant groups are unlikely to share. These and some other terms related to race and methodology are defined in a glossary (Table 1) appearing at the end of the manuscript.

By positing race to be a basis for pervasive and enduring racism rather than either non-salient in contemporary times or a cultural identity offering instrumental benefits, the core tenets of RFT and CRT stand in stark contrast to colorblind ideology as well as diversity orthodoxy. CRT and RFT have proliferated as frameworks for analyzing race-based inequalities in diverse contexts [32]. Moreover, use of CRT and RFT as methodological and epistemological frameworks for conducting research within IHEs has been growing [29, 34]. Notwithstanding this proliferation, a critical framing of racism as a systemic phenomenon with deep historical roots, is not typical of rhetoric invoked by university leadership in response to instances of racism on campus [13, 19, 27]. Instead, as argued above, institutional responses to local incidents of racism have generally been steeped in themes associated with colorblindness and diversity orthodoxy.

In addition to episodes involving school students or staff, it is becoming increasingly common for IHE administration to also release statements in response to racial injustice and violence in the nation more broadly [14–18]. The phenomenon of university-released statements became especially salient in the Summer of 2020, as the nation experienced the shockwaves of the video documentation of the murder of George Floyd. Since 2020, several scholars have investigated public reactions, especially those from universities and for-profit corporations, in

**Table 1. Glossary of significant race-related and methodological terms used in the manuscript.**

| Term | Definition |
|---|---|
| Ideology of Colorblind Racism/ Colorblind Racism/ Colorblindness/ Colorblind Ideology | Colorblind racism refers to individual and systemic discourses and practices that operate under the guise of race-neutrality. The ideology ignores or denies systemic and structural inequalities that continue to exist in society and contributes to perpetuating racial inequality through the use of race-neutral language and behavior. For a full account of the ideology of colorblind racism, refer to [1]. |
| Systemic Racism | Systemic racism refers to racial discrimination and differential treatment based on racial hierarchies embedded within social, political, economic, and cultural institutions in the United States. |
| Microaggressions | Microaggressions are subtle, brief, and commonplace acts or comments that convey derogatory messages and reinforce stereotypes towards marginalized individuals or groups [7]. Sue and colleagues [7] identify three main types of microaggressions: microassaults, microinsults, and microinvalidations. Microassaults involve explicitly discriminatory actions or remarks, such as racial slurs or overt exclusion. Microinsults, on the other hand, are subtle, indirect insults or demeaning messages that target a person's identity or background. They are often conveyed through dismissive comments or backhanded compliments, such as by asking a colleague of color how they got their job, implying that they received it through an affirmative action program rather than merit. Microinvalidations involve undermining or negating a person's experiences or identity, such as by denying the existence of systemic racism or dismissing someone's experiences of discrimination as being overly sensitive or exaggerated. For a full account, please see [7–8]. |
| Diversity Orthodoxy | An orthodoxy constitutes a set of widely shared ideas, beliefs, and practices that guide institutional discourse as well as policy, strategy, and action. Diversity orthodoxy refers to the institutionalization and legitimization of rhetoric and policies affirming the specific interpretation of 'diversity' as heterogeneity of persons based on a myriad of social and personal differences such as race, gender, ethnicity, nationality, and disability status by organizations. Diversity orthodoxy treats heterogeneity of persons as instrumentally beneficial for organizations. For a full account of Diversity Orthodoxy, refer to [26]. |
| Racial Formation Theory (RFT) | Racial Formation Theory is an analytical framework developed by Michael Omi and Howard Winant. The theory maintains that, rather than a fixed biological category, race is socially constructed and contingent on historical processes. Further, RFT emphasizes the role of individuals and communities in contesting racial categories and identities. RFT also suggests that contestation of racial classification has been and continues to be a site of political struggle. For a full account of Racial Formation Theory, refer to [4]. |
| Critical Race Theory (CRT) | Critical Race Theory (CRT) is a framework for examining the role of race and racism in society. According to CRT, race is a socially constructed and historically contingent category. Moreover, the theory maintains that racial hierarchies and racial discrimination are ubiquitous and deeply embedded within legal, social, political, economic, and cultural institutions, thereby affecting the opportunities and outcomes of racial groups. For details, see [31–32]. |

*(Continued)*

**Table 1.** (Continued)

| Term | Definition |
|---|---|
| Interest Convergence (CRT) | Interest convergence is the idea that significant racial progress, including legal gains, has only occurred in the United States when the interests of African Americans have coincided with the interests of the dominant white racial group. As such, despite their purported neutrality, institutions and laws serve the interests of dominant races and classes. Interest Convergence as a concept was coined by Derrick Bell. For details, see [36–37]. |
| Voice of Color Thesis (CRT) | The 'voice-of-color' thesis of CRT refers to the notion that experiences of discrimination accord marginalized groups with a 'competence' to speak about race and racism, which members of dominant groups are unlikely to share. For details, see [31–33]. |
| Prejudice | Prejudice, and specifically racial prejudice, refers to preconceived negative opinions or beliefs about a racial group. |
| Topic Modeling | Topic modeling is an automated procedure for locating themes or "topics" from a corpus of documents. The method draws on the notion that, rather than being absolute, meaning is inherently relational. In this case, relationality is measured through the co-occurrence of words in documents, which, in turn, are seen as 'bags of words.' For details, see [48–49]. |
| Two-Mode/Bipartite Network | A bipartite graph, $G = \{U, V, E\}$, is composed of two sets of nodes U and V and edges, E, that measure links between U and V. For details, see [50]. |
| Topic Investedness | The proportion of the statement composing a given topic. |
| Exponential Random Graph Models (ERGM) | ERGMs are statistical techniques for modeling networks. The ERGM framework assumes a stochastic environment in which edges are random variables and the number of nodes is fixed. |

the aftermath of this incident. These researchers have drawn on data from a variety of contexts including small pools of elite schools [15], broader representative samples of colleges and universities [14], as well as institutions that provide specialized training such as medicine or nursing [16–18]. Regardless of sample, these studies come to comparable conclusions about the ways in which racism is discussed in statements. Specifically, consistent with diversity orthodoxy, researchers find the themes of 'justice,' 'diversity,' and 'inclusion' to be featured prominently in IHEs rhetoric. In their analysis of statements released by 56 leading United States medical schools, Kiang and Tsai [16], for example, find that 40 use the term "inclusion," 33 use "diversity," and 29 use "justice." The authors also note that all institutions used some form of what they characterize as 'hopeful' language–rhetoric that invokes diversity as having positive instrumental value.

Second, consistent with colorblindness, researchers find that statements tend to avoid discussion of systemic racism. Veltman [14], for example, argues that schools rely on coded language that alludes to these themes rather than discuss them directly. Likewise, Brown et al. [17] analyzing statements issued by 35 medical schools and 10 national medicine-related organizations find that, while two-thirds of the statements mention the term "racism," only about half mention "systemic racism." More significantly, when racism is discussed, however, Brown et al. [17] find, it is generally framed in terms consistent with colorblind ideology—as an interpersonal and isolated phenomenon. Statements also generally avoided terms related to "whiteness" such as privilege and supremacy. As such, deep engagement with theories that treat racism as a systemic and historic phenomenon, including RFT and CRT, are largely missing from the discourse. Finally, scholarship on statements issued in the Summer of 2020 shows

that mentions of terms related to the police (such as "police officer" and "law enforcement") were mixed. Generally, researchers found that, although policing terms were mentioned often, statements do not centrally address policing. Knopf et al. [18], for example, note that nearly all statements in their sample included statements condemning police brutality, but "few statements emphasized that the killings were due to police violence" [18 p. 11] Veltman [14], likewise, finds that references to the police were in the context of universities discussing action steps to increase community trust in university police, while also affirming campus police as committed to serving and protecting the community.

Drawing on insights from these studies, our goal in this paper is to conduct a systematic investigation of statements issued by IHEs in the United States in the aftermath of George Floyd's murder. Our study contributes to this growing body of literature in several ways. First, we draw on a much larger sample (N = 356) than used by any study thus far. This larger-sized sample allows us to investigate the relationship between emergent statement themes and other IHEs' attributes such as geographic location, composition of student population, and prestige markers. We expect these variables to shape diversity rhetoric because academia in the United States is widely understood to be a status hierarchy such that those in positions of power exercise considerable control over academic practices and norms [38–44]. In this vein, research shows, for example, that strategies and practices tend to diffuse between IHEs, and that adoption is shaped by factors such as size, endowments, and rankings [45]. Second, the size of our dataset permits us to use quantitative techniques to analyze the data in a statistically rigorous manner. We use a machine-learning approach called topic modeling and a technique for the statistical analysis of networks called exponential random graph modeling (both described in detail in the Methods section) to locate themes in the statements as well as relationships between themes and other variables.

Finally, our study seeks to contribute to the literature investigating the evolution of the rhetoric on race and racism in the United States, especially in the context of higher education. While we analyze statements released at approximately one point in time, our objective is not limited to analyzing the rhetoric in that set of responses. We also aim to compare the dialogue invoked in the Summer of 2020 to findings from prior literature including other analyses of statements issued in the aftermath of George Floyd's death [9–20], which shows the dominance of colorblind ideology and diversity orthodoxy in dealing with issues related to race and racism in United States IHEs. Towards this goal, we view the murder of George Floyd as a watershed moment in the United States that has once again catapulted issues of systemic racism and police brutality to the forefront of American–and arguably global–public consciousness. Since then violence against other Black persons perpetrated by the police in the United States including Tyre Nichols in Tennessee and Irvo Otieno in Virginia and protests and institutional responses being issued thereafter, we believe that statements released in the Summer of 2020 are part of an ongoing and evolving conversation on racism and police violence in the United States. To the best of our knowledge, our study is the first comprehensive analysis of such statements issued by IHEs in the United States.

## Data

**Statements.** We focused on all IHEs included in the 'National Universities' rankings produced by U.S. News and World Report (USNWR) in 2021 (N = 388). USNWR started publishing evaluations of colleges and universities in the early eighties. Although rankings produced by the organization have received some criticism, USNWR has come to garner tremendous legitimacy as an evaluator of IHEs in the U.S. [46, 47]. Their evaluations are based on a variety of indicators of academic quality such as graduation rates, faculty and student resources, and

admissions selectivity. UNSWR draws on the Carnegie Classification for categorizing IHEs, a widely accepted standard in the U.S., to produce several distinct types of rankings. We draw on IHEs ranked in the 'National University' category, which includes colleges and universities that offer a range of undergraduate degrees, master's programs, as well as doctoral degrees. These schools are also at the forefront of academic research. As such, we exclude institutions focused primarily on undergraduate education such as liberal arts colleges, regional schools, and community colleges. Our primary reason for focusing on graduate-degree granting institutions is that much prior research on institutional responses has investigated such schools [e.g., 13–16]. Second, our goal of investigating the effect of rankings on shared themes is only feasible if institutions are ranked on the same evaluation system. Undergraduate schools, for example, are evaluated using different metrics owing to their distinctive organizational structure. Accordingly, it would be hard to reconcile and appropriately compare schools ranked across lists (such as Williams College, ranked highly in liberal arts schools, and Princeton University, ranked highly among National Universities).

Statements released by institutions ranked in the 'National Universities' list were located through keyword searches (including "George Floyd" and "President" or "George Floyd" and "statement" or "George Floyd" and "provost" or "George Floyd" and "chancellor") on IHE websites. Only statements made by heads of institutions such as the president, chancellor, or provost of the IHEs were used in this analysis. Thus, statements released by individual units within universities, for example, were disregarded. If an institution released multiple statements, only the first released statement was included in our sample. In instances where USNWR separately ranked universities with multiple campuses, like Rutgers University (for which three campuses are ranked), each campus was included as a unique entry in our dataset, if each listed school released its own statement. However, when multiple campuses of the same university were separately ranked, but a joint statement was released, we represented the campus in our dataset as a single school. This occurred only in two cases–University of Missouri and University of Michigan. In these two instances, we utilized the attributes for the highest ranked 'flagship' campus to represent the university system. When we could not find a statement on institutional webpages, we searched for statements on social media platforms such as Twitter and Facebook. Finally, statements posted in video or photo format were manually transcribed. This process yielded a total of 356 statements. Twenty-nine institutions ranked by USNWR did not release statements and were excluded from our analysis. On average, institutions in our sample released a statement one week after George Floyd's murder (mean = 7.24, standard deviation = 4.26), with the first statements being released two days after the murder and the last statement, forty-two days after. The statements also varied considerably in length ranging from a minimum of three sentences to a maximum of eighty-five.

**Attributes.**   In addition to the statements, we also collected data on a variety of institutional attributes, which we describe next. Data descriptives are shown in Table 2.

- Rankings (continuous variable): IHE rankings were sourced from USNWR National Universities rankings released in 2021. In this year, USNWR ranked schools in the range of 1–296 and the remaining schools were rated as a range '297–389.' We code the bottom range as having a rank of 297.

- Black undergraduate student percentage (continuous variable): We drew on data from 'College Factual' to determine the percentage of the undergraduate student body identifying as Black and/or African American (minimum = 0 percent; maximum = 94.8 percent). College Factual uses data from the Integrated Postsecondary Education Data System (IPEDS), and specifically the "EFA" (Exploratory Factor Analysis) dataset to determine demographic data

**Table 2. Descriptive statistics for data.**

|  | Mean | Median | Standard Deviation |
|---|---|---|---|
| Black Student Percentage | 10.39 | 6.5 | 13.7333 |
| Female Student Percentage | 56.33 | 56.2 | 9.53 |
| Statement Length (word count) | 508.242 | 439.5 | 316.911 |
| Statement Length (sentence count) | 19.971 | 17.0 | 12.014 |
| Time of Statement Release since May 25, 2020 (in days) | 7.24 | 7 | 4.2254 |
| Geographic Region | N | Proportion of sample |  |
| Northeast | 84 | .2360 |  |
| South | 144 | .4045 |  |
| West | 58 | .1629 |  |
| Midwest | 67 | .1882 |  |
| Pacific | 3 | .0084 |  |
| State Political Affiliation |  |  |  |
| Blue | 160 | .4494 |  |
| Red | 150 | .4213 |  |
| Swing | 46 | .1292 |  |
| Flagship Status |  |  |  |
| Flagship | 49 | .1376 |  |
| Not Flagship | 307 | .8624 |  |
| HBCU Status |  |  |  |
| HBCU | 9 | .0253 |  |
| Not HBCU | 347 | .9747 |  |

on students enrolled in 4-year universities, including race, gender, attendance status, and student level. College factual data for this and other variables were accessed in August 2021.

- Female undergraduate student percentage (continuous variable): We collected data on the percentage of undergraduate students that are female from College Factual (minimum = 2.4 percent; maximum = 93.9 percent).

- Geographic region (categorical variable): We used five United States census designations–Northeast, South, Midwest, West, and Pacific (Hawaii and Alaska)–to code institutional geographic location.

- State political affiliation (categorical variable): We coded the political affiliation of the state in which institutions are located based on the results of the 2016 and 2020 United States Presidential elections. States that voted Republican or Democrat in both elections were coded as "red" and "blue" respectively. States that voted differently in the two elections were considered "swing" states.

- Flagship status (binary variable): Flagship status was determined using a list from College Raptor. The organization defines a flagship school as 'the most prominent university' in each state, which receives the greatest amount of state funding. Our sample includes 49 flagship universities.

- Historically Black College and University (HBCU) status (binary variable): Historically Black colleges and universities status was determined from the website, 'The Hundred Seven,' which compiles information about the 107 HBCUs in the United States. Our sample includes nine HBCUs.

- Time (continuous variable): We calculated the difference between the release of a statement and the number of days since the murder of George Floyd (May 25, 2020) as a continuous variable. We could not locate time stamps for eight statements in our sample. We used the highest number–forty-two days–for those statements. We also tried the mean as well as median number of days in place of the maximum. The results of the estimation remained unchanged regardless of the value used for the missing data.

## Method

### Topic modeling

We use the techniques of topic modeling and exponential random graph modeling to analyze the data. Topic modeling is an automated procedure for locating themes or "topics" from a corpus of documents. The method draws on the notion that, rather than being absolute, meaning is inherently relational. In this case, relationality is measured through the co-occurrence of words in documents, which, in turn, are seen as 'bags of words.' A topic is a set of words that tend to occur together within the corpus more often than by chance. As such, each topic is a distribution of words, and each document is composed of a set of topics. The order of the words as well as other parts of language including syntax is considered irrelevant to the process. We deploy a commonly used technique called Latent Dirichlet Allocation [48] implemented in a tool called Mallet [49] to generate the topic models. Our corpus meets the basic assumptions of Latent Dirichlet Allocation that statements are a distribution of topics, and topics are a distribution of words where word-order is irrelevant. Moreover, the documents in our corpus generally have a large number of words (mean = 508.2).

The topics so generated are not meaningful in themselves but need to be interpreted by an analyst accounting for the broader context in which the corpus arose. We fit many models with the number of topics ranging from ten to thirty. The first two co-authors independently analyzed each model with the goal of locating one that made most sense in the context of the data. We found models with fewer than fifteen topics to be lacking in exhaustiveness. Likewise, models with greater than twenty-four topics had too much thematic overlap, leading some topics to be indistinguishable from others. We narrowed down to a smaller subset, from which we chose a model with eighteen topics. This choice was supported by the coherence score. We also ran the log-likelihood associated with each model. Among models with at least fourteen topics, our preferred solution with eighteen topics had the lowest log likelihood. Only models with ten or twelve topics had slightly lower scores, but, as stated above, we found those solutions to be substantively inadequate.

We applied several techniques to pre-process the corpus before generating the topics. First, we 'tokenized' the text in our corpus by converting the statements to 'bags of words.' We then converted all tokens to lowercase words such as 'university,' 'racism,' and 'solidarity,' Second, we removed tokens that are typically considered extraneous to the modeling process including: symbols, web URLs, punctuation marks, and stop-words (such as articles) based on a standard pre-compiled list (N = 595). We also eliminated IHE-specific salutations such as 'professor'. Third, we treated several sequences of pairs (bigrams) or triplets (trigrams) of words appearing together such as 'systematic racism,' 'african american,' and 'george floyd death' as single tokens. Lastly, we filtered out tokens that occur in more than seventy percent of the statements as well as those that occur less than five times. At the higher end, the filtering is useful for removing noisy tokens that occur in most statements, and hence are unsuitable for detecting patterns. Filtering on the lower end is necessary to eliminate statement-specific details such as

school names (e.g., UC System, UMass System). The first and fourth listed authors, nevertheless, reviewed all tokens appearing five or fewer times with the goal of including any that were important for detecting themes based on a qualitative analysis of fifty statements.

## ERGM

In addition to the distribution of words per topic, LDA also produces a distribution of topics over documents. We used this matrix to create a two-mode network or bipartite graph [50]. A bipartite graph, $G = \{U, V, E\}$, is composed of two sets of nodes U and V and edges, E, that measure links between U and V. In our case, the two-mode network consists of topics and statements. An edge in this network denotes the proportion of the statement being composed of a given topic. We refer to this also as the degree of a statement's 'investedness' in a topic. A university or college is more invested in a topic, for example, when a higher proportion of its statement is devoted to that topic. We deduce this from the document-topic probabilities vector or topic mixture, which shows the estimated proportion of words from a given statement that are generated from all topics. The sum of proportions across all topics totals one for a given institution's statement. Following Curran et al. [51] and Vlegels and Daenekindt [52], we dichotomize these edges by coding a tie as having a value of '1' if the proportion is at least twice as high as would be the case if topics were uniformly distributed across statements. As our chosen solution has 18 topics, this means that we coded an edge as 1 if the proportion was at least ((1/18)*2) or 0.1111. The remaining ties were coded as 0.

This process yielded a binary two-mode network with a density of 6.8 percent, meaning that more than four-fifths of the ties in the original topic-statement matrix were less than 0.11. Two-mode networks can be projected to yield two one-mode networks comprising links between nodes of the same subset. The procedure involves multiplying the matrix by its transpose or vice versa. We used this procedure to generate a one-mode projection comprising links between statements. Two statements are linked if they share at least one topic in common (based on the 0.1111 cutoff described above). The resultant matrix has a density of 0.085: a little over ninety percent of the statements have no topics in common with others. Among those that are connected, less than three percent share two topics in common; the remaining share only one topic in common. We dichotomize this one-mode matrix and analyze it statistically using ERGM.

The ERGM framework assumes a stochastic environment in which edges are random variables and the number of nodes is fixed. Two types of variables are typically used in ERGMs. First, endogenous variables such as edges, stars, and shared partners are conceptualized as microstructures that concatenate to produce the observed network. These configurations are theorized to be self-organized structural tendencies where network ties are probabilistically generated out of the existence of other ties. We do not use these types of variables in our analysis. This is because one-mode projections are known to be highly dense, so modeling endogenous structural features using the ERGM framework is less interesting. The process is also less feasible because projected networks often produce degenerate distributions (where all or most of the probability distribution is clustered around a few possibilities, most notably the full or near-full graph) (see, [53]). Instead, we focus on 'exogenous' variables, described next, that are substantively important to our research agenda.

Exogenous attribute variables test if attributes of nodes are associated with the formation of ties. Two types of effects are often used in the modeling process. First, homophily is the tendency for similar nodes to be connected to each other [54]. We can use homophily variables to test if statements issued by IHEs that are similar along attributes such as percentage of Black student population, prestige rankings, and geographic location are more likely to be connected

through shared themes. Likewise, differential connectedness variables test if IHEs with specific attributes (such as high rank) are more likely to be connected in the network.

The exponential family of distributions applied to network data is characterized by the following equations:

$$P_\theta\{Y = y\} = exp(\theta'u(y) - \varphi(\theta)) \tag{1}$$

$$exp(\varphi(\theta)) = \sum_y exp(\theta'z(y)) \tag{2}$$

where θ is the vector of parameters to be estimated, u(y) is any vector of sufficient statistics, endogenous and exogenous, and φ(θ) is a normalizing constant that ensures the probability distribution in Eq (1) is proper. Models are fit using Monte Carlo Markov Chain Maximum Likelihood Estimation (MCMCMLE).

ERGM is increasingly being used in two ways. Traditionally, the goal of fitting an ERGM has been to find the best possible model with the goal of replicating the structure of the empirical network. More recently, ERGM is also being used to test hypotheses without necessarily focusing on locating the best possible fit for the data. We use the latter approach in this paper.

## Analysis

### Topic model analysis

Table 3 shows results of the topic modeling. As discussed above, after exploring many options, we decided on a model with eighteen topics. The names of the topics, based on an in-depth analysis of their content, are shown in the first column of the table. The second column shows a list of the most frequently occurring tokens in that topic followed by a brief description in the subsequent column. Each topic description is also accompanied by an example from statements that reflect the ethos of the content associated with the topic. Finally, the table also lists five broad domains in boldface–'Racism and Racial Violence and Injustice,' 'Institutional Reckoning and Response,' 'Rhetoric on Race as a Historical Social Problem,' 'Christian and Humanist Values,' and 'COVID-19'–that we used to classify the topics. The domains and, hence, topics are arranged in decreasing order from most to least related to what we identified as 'race-centric' issues. These are issues that we considered to be explicitly focused on race in the United States. Topics with a high concentration of such issues contained several explicitly race-related tokens in their top keywords such as 'black,' 'discrimination,' 'systemic racism,' and 'equity'. Topics with low prevalence of such issues contained almost no such tokens. The top-fifteen tokens in the second topic ('Safe Return to Campus') of Domain 5 ('Covid-19'), for example, contain almost no tokens that we consider to be expressly race-related issues. We do not re-summarize the topics here, as those details are available in Table 3. Instead, we use this section to discuss the five domains and how topics are linked within those domains.

The first domain, 'Racism and Racial Violence and Injustice,' comprising twenty-nine percent of the total corpus, includes topics that are most clearly focused on issues of contemporary racism in the United States. Racial injustice, seen through the lens of numerous violent and deadly incidents explicitly referenced and discussed in the statements, is a common theme in this domain. Names of victims appear frequently and the term, 'systemic racism,' is recurrent. In contrast to prior literature which demonstrates the proliferation of colorblindness and diversity orthodoxy in higher education rhetoric, topics in this domain, we find, resonate strongly with the tenets of CRT, and especially the notion that systemic racism is deeply embedded within United States' social, political, and economic institutions. The first topic, 'Racial Injustice,' notably, draws the readers' attention to this widespread nature of racism and

**Table 3. Topic model labels, top 15 tokens, and descriptions.**

| Topic Label | Top 15 Tokens | Topic Description (This topic refers to…) |
|---|---|---|
| **Domain 1: Racism and Racial Violence/Injustice** | | |
| (1) Racial Injustice | injustice; stand; respect; violence; act; hate; discrimination; witness; family; treat; bring; compassion; solidarity; hatred | Racial injustice in the form of hate and discrimination writ large. |
| "Racially motivated injustices and tragedies in Georgia, Minneapolis, New York's Central Park and elsewhere have once again brought hatred and violence against African Americans to the forefront of our collective consciousness. These incidents are disturbing and reprehensible." -University of Missouri | | |
| (2) Racial Violence and Black Lives Matter | kill; commitment; black; acknowledge; systemic racism; live; recognize; pain; breonna taylor; work; inquiry; matter; equity; victim | Mentions of racial violence against Black lives and includes the names of victims of racial violence. |
| "Just three weeks ago I wrote on Twitter about the horrific shooting of Ahmaud Arbery, who was shot dead while jogging in the coastal city of Brunswick, Georgia, in late February. In mid-March, Breonna Taylor, a young emergency medical technician in Louisville was killed in her apartment when police entered. And on Monday of this week George Floyd was killed in Minneapolis when an arresting officer kneeled on his throat for over eight minutes."—Rice University | | |
| (3) Racial Police Brutality | police; justice; america; call; mr floyd; officer; die; freedom; share; law; act; man; child; african american | Themes of racial violence pertaining to police brutality against African Americans/Black individuals in the United States. The topic includes explicit mentions of Mr. George Floyd as a victim of police brutality. The topic also includes mentions of police as explicit perpetrators of violence. |
| "George Floyd died one week ago today, handcuffed and pinned to the ground by Officer Derek Chauvin of the Minneapolis Police Department. The video of the arrest shows Officer Chauvin with his knee on Mr. Floyd's neck while Mr. Floyd pleads with the officer, telling him that he is in pain and that he cannot breathe, before he calls out for his mother. Officer Chauvin kept his knee on Mr. Floyd's neck for more than two minutes after Mr. Floyd became non-responsive."—Drake University | | |
| (4) Death and Victims of Racial Violence | death; violence; speak; nation; member; minneapolis; country; university community; commitment; condemn; live; mourn; city; racism | Descriptions of death and the condemnation of death (passive) attributable to racial violence, without explicit mention of law enforcement. |
| "We are not together in person, but we must rise with one united voice to call out and condemn the racism and targeted racial violence happening nationwide where senseless acts of excessive force and aggression have resulted in death, fear and suffering."-University of Massachusetts, Lowell | | |
| (5) Student Support on Racial and Social Justice | student; create; work; impact; support; focus; address; statement; member; mission; educate; commit; opportunity; issue | Themes within university statements focused on student support, explicitly pertaining to issues of social and racial justice. |
| "Unfortunately, our campus is not immune from such pernicious forces. We must recognize the stereotyping, stigmatization and marginalization of diverse individuals and communities that occur on our own campus and work to tackle them. We have made some progress in the past several years through our IDEAL initiative, overseen by Provost Drell, but we need to do more and act with even greater urgency to create an inclusive, accessible, diverse and equitable university for all our members. And we need to start now, including working to eliminate the anti-Black racism that has been laid bare by the events of the past weeks."-Stanford University | | |
| **Domain 2: Institutional Reckoning and Response** | | |
| (1) University Diversity, Equity, and Inclusion | inclusion; country; diversity; president; continue; responsibility; join; opportunity; dialog; diversity equity; embrace; exist; action; pledge | Themes within university statements pledging support for Diversity, Equity, and Inclusion policies and practices. |
| "As a remarkable and positive community of enlightened individuals, we are unequivocally committed to diversity, equity and inclusion. We believe that every person is worthy of dignity, care, respect, compassion and opportunity. We know that no one should be judged, helped or hurt because of their skin color, gender identity, ethnicity, religion, ability or sexuality. Individuality is valued and celebrated at Adelphi." -Adelphi University | | |
| (2) Institutional Action Through Dialog | university; action; week; lead; conversation; step; time; hold; bring; race; leadership; clear; group; open | IHEs stated commitment to creating spaces within the campus for dialog on issues of race, racism, and discrimination. |
| "Unification starts with listening, communicating and understanding. We can begin with an open and transparent dialog. This is critical if we are going to make any progress.<br>That dialog can start with a goal of better understanding how we each experience the world differently from each other. As a university, we will pursue these actions and, from these, learn of other ways we can effect change and play a role in moving toward greater unity:…" -University of Phoenix | | |
| (3) Institutional Commitment to Listening and Learning | institution; commit; work; leader; experience; individual; hear; forward; listen; campus; learn; feel; force; result | IHEs stated commitment to listening to and learning from campus community members, without any explicit mention of race. |
| "We do this by mourning with others, by being uncomfortable listening to their pains instead of trying to explain it away or instead of telling them how they should feel or instead of jumping to easy answers."- Biola University | | |
| (4) Institutional Action through Education and Research | effort; include; experience; university; serve; work; faculty; provide; education; program; college; share; process; national | IHEs stated commitment to facilitating educational and research efforts, without explicit mention of race or racism. |
| "As an educational institution, we solve problems through the myriad efforts of our faculty, staff and students. They are engaged in research, teaching and service to dismantle racist policies, such as those that result in funding Pennsylvania's public schools in a way that disadvantages black children. As we continue to advocate for an equitable funding system that guarantees the same quality public education for all school children regardless of ZIP code, Temple will continue to stand in the gap."-Temple University | | |

*(Continued)*

**Table 3.** (Continued)

| Topic Label | Top 15 Tokens | Topic Description (This topic refers to…) |
|---|---|---|
| (5) Inclusive Environment on Campus | color; people; university; staff; students; faculty; occur; racism; event; protect; live; understand; build; ensure; require | Stated university commitment to creating an inclusive, safe, and non-racist campus environment, especially for people of color within the university. |

"To our students, staff and faculty of color–I see you. I hear you. Given these tragic incidents and mounting tension building in our own city and across the country, I know many of you are in deep pain having to confront these inequities, sometimes on a daily basis. Please know that I am here to support you, this university is here to support you and we will continue our endeavor to provide an environment where everyone can thrive." -University of Louisville

| Topic Label | Top 15 Tokens | Topic Description |
|---|---|---|
| (6) Campus Resources for Diversity | support; event; care; reach; service; difficult; member; diversity inclusion; office; encourage; campus community; resource; feel; center | Discussions of existing campus resources for diversity and inclusion as expressed within IHE's statements during the Summer of 2020. |

"I want to remind our campus community of resources that are available to assist you. I encourage anyone who needs it to reach out to these campus resources to provide you with support, compassion, and understanding."-Wright State University

### Domain 3: Rhetoric on Race as a Historical Social Problem

| Topic Label | Top 15 Tokens | Topic Description |
|---|---|---|
| (1) "National Historical Moment" | protest; history; watch; moment; city; individual; human; answer; nation; give; law enforcement; humanity; remember; point | Discussions within statements on the relation between the protests during the Summer of 2020 to the United States' racial history, particularly as it pertains to communities' relations with law enforcement. |

"First, there must be justice for George Floyd, and it is clear that public attention has brought needed scrutiny into the judicial process, just as it did when videos of the shooting of Ahmaud Arbery in Georgia gained public attention. The world is watching."—Russell Sage College

| Topic Label | Top 15 Tokens | Topic Description |
|---|---|---|
| (2) Twin National Problems | country; society; justice; work; long; people; power; injustice; student; continue; address; solution; inequality; fact | Rhetoric within IHEs' statements on the injustice and inequality marking the two national problems of the Summer of 2020, police brutality against Black people, as well as the COVID-19 pandemic. |

"The past several months have presented unprecedented challenges for our community, the nation, and the world. Events of the past week have reminded us that while we are all focused on keeping our loved ones safe and healthy, the underlying inequities within our society remain. In fact, we must acknowledge that societal inequalities are actually being exacerbated by the COVID-19 pandemic. The protests of the past few days, ignited by the killing of George Floyd, but truly fueled by the continued targeting, demonization, and abuse of black people across our country, highlight legitimate anger, which I share." -Worcester Polytechnic Institute

| Topic Label | Top 15 Tokens | Topic Description |
|---|---|---|
| (3) National Legacy of Racism | change; racism; world; society; place; seek; nation; confront; reality; form; alumnus; face; resolve; continue | Discussions within statements on the national legacy of racism, especially against African Americans, in the United States. This topic within statements also emphasizes the need to acknowledge and confront the racial realities of the past and present. |

"In the midst of this devastating experience, the original fault line of our republic has been exposed once again for the nation. We grieve the killing of George Floyd in Minnesota, Breonna Taylor in Kentucky, and Ahmaud Arbery in Georgia as unconscionable acts of violence. Their deaths, and subsequent nationwide protests, once again present our country—and each one of us—with the imperative to confront the enduring legacy of slavery and segregation in America." -Georgetown University

| Topic Label | Top 15 Tokens | Topic Description |
|---|---|---|
| (4) Space, Time, and Emotions | nation; day; word; voice; issue; face; concern; good; hurt; reflect; year; month; family; show; strength | Diffuse references to affect and emotions, such as hurt and pain, within a particular space and time in the United States. It rhetorically references the nation and year as loci of the affects. |

"It is hard to find words to express the collective hurt, anger, and shame our nation is feeling. It's particularly hard as we've been here before and we haven't realized the needed change. We haven't put racial inequity behind us. We have failed to right the wrongs built up over centuries." -Rochester Institute of Technology

### Domain 4: Christian and Humanist Values

| Topic Label | Top 15 Tokens | Topic Description |
|---|---|---|
| (1) Christian and Humanist Values | love; heart; god; people; live; world; stand; pray; justice; peace; brother; sister; african american; life | References a commitment to Christian and humanist values such as love, peace, and dignity, and makes explicit references to God, Christ, and sin. |

"As a follower of Jesus Christ, I turn first to Scripture to assess my feelings and interpret the situation. I have come to reaffirm some deeply held theological convictions–
• We live in a fallen and broken world
• The only real reconciliation is Gospel reconciliation
• Every person of every ethnicity was created in the image of God
• Racism in all forms is anti-Gospel
• The life of every person is precious and valuable to God"
-Carson-Newman University

### Domain 5: COVID-19

| Topic Label | Top 15 Tokens | Topic Description |
|---|---|---|
| (1) COVID-19, Loss, and Grief | life; let us; fear; challenge; pandemic; covid; share; truth; sense; loss; video; lose; suffer; crisis | The impact of COVID-19 including references to crisis, a sense of fear, and feelings of loss and grief suffered as a result of the COVID-19 pandemic. |

"There is of course great sadness sweeping across our world and our country because of the pandemic. Many have lost friends and family members. Many more are living in isolation and under constant threat of danger to their health. Tens of millions have lost their jobs and collectively our country is facing a level of unemployment and financial distress we have not seen in 90 years." -Rice University

*(Continued)*

**Table 3.** (Continued)

| Topic Label | Top 15 Tokens | Topic Description (This topic refers to. . .) |
|---|---|---|
| (2) Safe Return to Campus | campus; plan; student; work; fall; state; health; return; faculty staff; continue; safe; member; learn; provide | University plans for COVID-19 mitigation measures as well as campus reentry and safety plans for the Fall of 2020 following the COVID-19 outbreak in the United States. |

"It is clear from our new and returning students and other stakeholders that there is a strong desire to return to campus in the fall. Our plans have been developed with the dual goals of: (i) supporting a safe and healthy campus environment for our community and, (ii) seeking to provide the best education and living experience possible for our students under these most challenging circumstances. . ." -Stevens Institute of Technology

violence against Black people in the United States. The second, third, and fourth topics are similar, but also contain uniquely differentiable elements. The second topic, 'Racial Violence and Black Lives Matter (BLM),' emphasizes BLM as a significant social movement engaging in political struggle against racial injustice [4]. Significantly, the trigram "Black Lives Matter," occurs twenty-four times in the corpus, and all those occurrences are captured within this topic. The fourth topic, 'Death and Victims of Racial Violence,' condemns violence against Black people and invokes calls for action but draws on passive language and does not name perpetrators of violence. The term 'police,' for example, does not appear at all in the topic. Likewise, while the term 'kill' is not prominent, 'death' occurs most frequently.

The third topic, 'Racial Police Brutality,' focuses on the role of the police as perpetrators of violence in incidents of racism. However, rather than focusing on individual prejudices or biases as pervasive in rhetoric consistent with colorblindness, this topic draws attention to racial discrimination in the criminal legal system, emphasizing 'interest convergence,' one of the core tenets of CRT, which holds that the law is fundamentally tilted in favor of racially dominant groups. The term "police" appears nearly exclusively in this topic. Notwithstanding, we find that while there are many references in statements to the tense relationship between law enforcement and communities of color in the United States, there were few explicit references to 'police brutality.' The token appears all of 9 times in 8 statements. In contrast, 'George Floyd' appears 298 times in 198 statements, and 'diversity' appears 219 times in 115 statements. An excerpt from Oregon State University showcases one of the few instances when the police are directly named as actors.

"The primary role of police in America is to provide for the safety of all people by protecting them from criminals and to hold each of us accountable to the law. We expect police to apprehend criminals and work within the legal system to make certain that justice is blind and all are held accountable to the law. We all have watched in horror videos being replayed over the past week showing the life of George Floyd brutally taken from him by a white police officer in Minneapolis, Minn., while three other officers sworn to uphold the law looked on in indifference. The officer who killed Mr. Floyd was arrested and all four of the officers were fired, but the other three officers simply went home. Sadly, this horrific event is just the latest in a seemingly endless stream of acts of violence against Black and other people of color by police who are sworn to protect and serve them."

–Oregon State University

Further analysis (discussed in greater detail below) shows that IHEs that contain high percentages of Black undergraduates are more likely to be invested in the topic we label 'Racial Police Brutality.' Likewise, IHEs located in 'blue' states that voted Democratic in the 2016 and 2020 presidential elections are also more likely to draw on this topic as compared to those located in 'red' states that voted Republican. The final topic classified within this domain,

'Student Support on Racial and Social Justice,' shifts gears towards student support on campus associated with issues of racial justice. Significantly, as exemplified by the Stanford University statement, this topic depicts racial (in)justice on campus as yet another instance of broader marginalization of people of color, neither a school-centric nor an isolated event. This is a considerable departure from findings based on prior analyses, which shows that university leaders tended to frame racist incidents on campus as aberrant and detached from broader societal issues.

This final topic in the first domain segues well into the second domain, 'Institutional Response and Reckoning,' which pivots away from explicit discussions of racial injustice and violence towards university and college sentiments and actions. This domain comprises about a third of the total corpus, and the topics within this domain are reflective of and consistent with rhetoric associated with diversity orthodoxy in IHEs. Specifically, we find two broad themes in this domain. First, topics emphasize IHEs' commitments to diversity, equity, and inclusion on campus. Here, on the one hand, diversity is framed as a matter of cultural difference, something to be celebrated (Topic 1: 'University Diversity, Equity, and Inclusion'). On the other hand, diversity is seen through the lens of marginalized racial groups whose members are experiencing pain and discomfort in the current climate (Topic 3: 'Institutional Commitment to Listening and Learning' and Topic 5: 'Inclusive Environment on Campus'). This latter framing, evident in the statement issued by Biola University (see, Table 3), also reflects the centering of marginalized voices or the voices-of-color thesis associated with CRT. It is worth noting that, while race is modal, diversity, in these topics, is also used to signal difference based on other attributes such as gender and religion. This is consistent with findings from the diversity orthodoxy literature [24, 26]. In the second broad theme, IHEs propose the creation of spaces on campus to facilitate dialogue and commit resources towards research and teaching initiatives aimed at acknowledging and addressing issues of racism (Topic 2: 'Institutional Action Through Dialog,' Topic 4: 'Institutional Action through Education and Research,' and Topic 6: 'Campus Resources for Diversity'). This finding mirrors Iverson's [22] democracy frame within diversity orthodoxy–calls to action for change based on a shared commitment to equity and inclusion.

The third domain, 'Rhetoric on Race as a Historical Social Problem,' is composed of topics that frame contemporary racism as a social problem in the United States with deep historical roots. Topics within this domain, comprising twenty-two percent of the corpus, significantly draw on RFT as a framework for confronting the differential worth assigned to African Americans, as a historical process in the United States. Notably, situating present-day racism within racialized historical processes in the United States departs from prior IHEs rhetoric typically leveraging either colorblind or diversity-based race frames. Tokens that are referents to time such as 'history,' 'remember,' 'long,' 'month,' and 'moment' are recurrent themes in these topics. Excerpts from Georgetown University and Rochester Institute of Technology showcase emphasis on the United States' legacy of slavery and racism, "the original fault line," which is seen as continuing to shape the life chances of African Americans today. The topics also emphasize political conflict and collective action as necessary and legitimate for producing social change, another theme associated with RFT, which maintains that institutionalized racial discrimination continues to be a domain for social movements and political struggle in the United States. Much like time, tokens associated with activism such as 'protest,' 'power,' 'injustice,' 'change,' and 'confront' occur frequently. This domain also includes a topic (Topic 2: 'Twin National Problems') that combines inequalities manifest in the COVID-19 pandemic and racial inequality as a "twin" problem plaguing the United States, further reflecting the CRT tenet that systemic racism is reflected in multiple domains of American life.

The fourth and final topic in this set, 'Space, Time, and Emotions,' which contains themes surrounding emotions associated with the United States legacy of racial violence and inequality, transitions into the next domain comprising a single topic and five percent of the corpus–'Christian and Humanist Values.' As suggested by the title, the topic, highly present in statements issued by IHEs with religious affiliations, invokes themes of compassion and religion to make sense of racial injustice and inequalities. This topic draws on colorblind race frames by shifting away from considerations of racism as a social problem, focusing instead on the imperative to engage with others' humanity, regardless of race, as both a civic and moral duty. Significantly, however, this domain is composed of a single topic and the standard deviation of statement investedness in this topic is higher than any other in the corpus. Greater than forty percent of Carson-Newman University and Biola University, for example, are dedicated to this topic. In contrast, less than one percent of statements issued by Stanford University and Stevens Institute of Technology are focused on it. The implication is that while some IHEs are deeply invested in these themes, most refrain from invoking them at all.

The final domain, 'COVID-19,' comprising two topics ('COVID-19, Loss, and Grief' and 'Safe Return to Campus') and about a tenth of the entire corpus, pertains to themes related to the COVID-19 pandemic. In contrast to the 'Twin National Problems,' topic within the third domain, 'Rhetoric on Race as a Historical Social Problem,' topics under this final fifth domain contain few references to racism. Instead, they capture sections of statements dedicated to the discussion of the effects of COVID-19 such as feelings of isolation, implications of job loss, and the safe and healthy return of students and faculty to campus.

Overall, our findings are suggestive of considerable variation in rhetoric used in these statements, with some topic domains remaining consistent with diversity orthodoxy rhetoric, and others assuming more critical discourse reflective of the CRT and RFT frameworks. While IHEs in our dataset often make implicit references to race and racism in the United States, a large proportion also contain themes that confront issues of racial violence and police brutality explicitly, using terms like "murder," "kill," "violence," and "discrimination". Likewise, statements also display variation in framing racism as both an interpersonal and systemic social problem. This contrast is evident in comparing sections of statements, such as the University of Missouri statement, with others like Oregon State University and Georgetown University. Whereas Missouri highlighted the importance of "personal responsibility and action to provide respect and caring for others in all of our interactions," the latter two schools focused on the systemic nature of racism, emphasizing "the enduring legacy of slavery and segregation in America" (Georgetown University Statement, 2020). These findings contrast with much prior work, which shows that IHEs' discourse on race is generally focused on diversity and limited to viewing racism as a matter of personal prejudice.

While this analysis offers an overview of the landscape of the themes invoked in university statements, it offers little leverage on how rhetoric varies based on IHE attributes and location. We investigate these connections in the next section.

## Attributional analysis

Table 4 shows results of the ERGM model, which tests the statistical significance of nodes to be connected to each other based on attributes described above. As the network we model is a one-mode projection of a two-mode network, we do not model structural features of the network. Instead, we investigate if shared themes are more likely if IHEs issuing them share salient attributes such as HBCU or flagship status. Likewise, we test if positionality on continuous variables such as rankings and percentage undergraduates that are female is associated with more connectedness. Fig 1 depicts bar graphs showing the distribution of the five major

**Table 4. ERGM results.**

| Effect | Estimate | Standard Error |
|---|---|---|
| Edge | -2.6292* | 0.133 |
| Religious Affiliation Activity | -0.1475* | 0.033 |
| Religious Affiliation Homophily | 1.4431* | 0.063 |
| Flagship University Activity | 0.0722 | 0.038 |
| Flagship University Homophily | -0.15 | 0.129 |
| HBCU Activity | 0.9236* | 0.139 |
| HBCU Homophily | -0.0388 | 0.512 |
| Rankings Activity | -0.0009* | 0.000 |
| Rankings Homophily | -0.0005* | 0.000 |
| Undergraduate Percentage Black Activity | 0.0019 | 0.002 |
| Undergraduate Percentage Black Homophily | -0.0082* | 0.003 |
| Time Activity | 0.0061 | 0.004 |
| Time Homophily | -0.0037 | 0.004 |
| Undergraduate Percentage Female Activity | 0.0026* | 0.001 |
| Undergraduate Percentage Female Homophily | 0.0090* | 0.002 |
| Region Homophily | -0.0974* | 0.034 |
| Political Affiliation Homophily | 0.1554* | 0.030 |

* Estimates are considered to be statistically significant if they are more than twice as large as the associated standard error.

topic domains from Table 3 by select attributes. These distributions help to offer context for the statistical findings from the ERGM models.

Generally speaking, positive parameter estimates in ERGMs suggest that configurations occur more often than by chance, having accounted for all other variables included in the model. Thus, a positive estimate for homophily by political affiliation suggests that connectedness between statements issued by IHEs that are in states that voted similarly on recent Presidential elections is more frequent than one might expect based on other variables included in the model. Likewise, a negative parameter estimate is indicative of configurations occurring less often than expected. There is one exception to this general rule: homophily on continuous variables measures likelihood of connectedness based on distance between values. Thus, homophily on rankings, for example, measures if statements issued by IHEs are more likely to be connected if the rank-distance between those IHEs is smaller. Accordingly, a *negative* estimate is indicative of homophily in the case of variables measured continuously. Estimates in the ERGM framework are considered significant if the reported standard error is less than half the corresponding parameter estimate. The models are fit using PNet [55].

The estimate for the 'edge' parameter, a measure of the baseline propensity for tie formation, is similar to the intercept in a linear regression. 'Activity' parameters capture the tendency for IHEs fitting those attributes to be more connected to others by virtue of shared themes. The results show that religiously affiliated IHEs are less likely to be connected to others, but more likely to be densely tied to other schools like themselves. Fig 1 offers confirmation and context for this finding: religiously affiliated IHEs are significantly less likely to invoke topics related to 'Racism and Racial Violence,' as well as 'Institutional Reckoning and Response,' but considerably more likely to use 'Christian and Humanist Values' in their statements. The implication is that religiously affiliated IHEs are more likely than others to draw on themes consistent with colorblind ideology in this corpus.

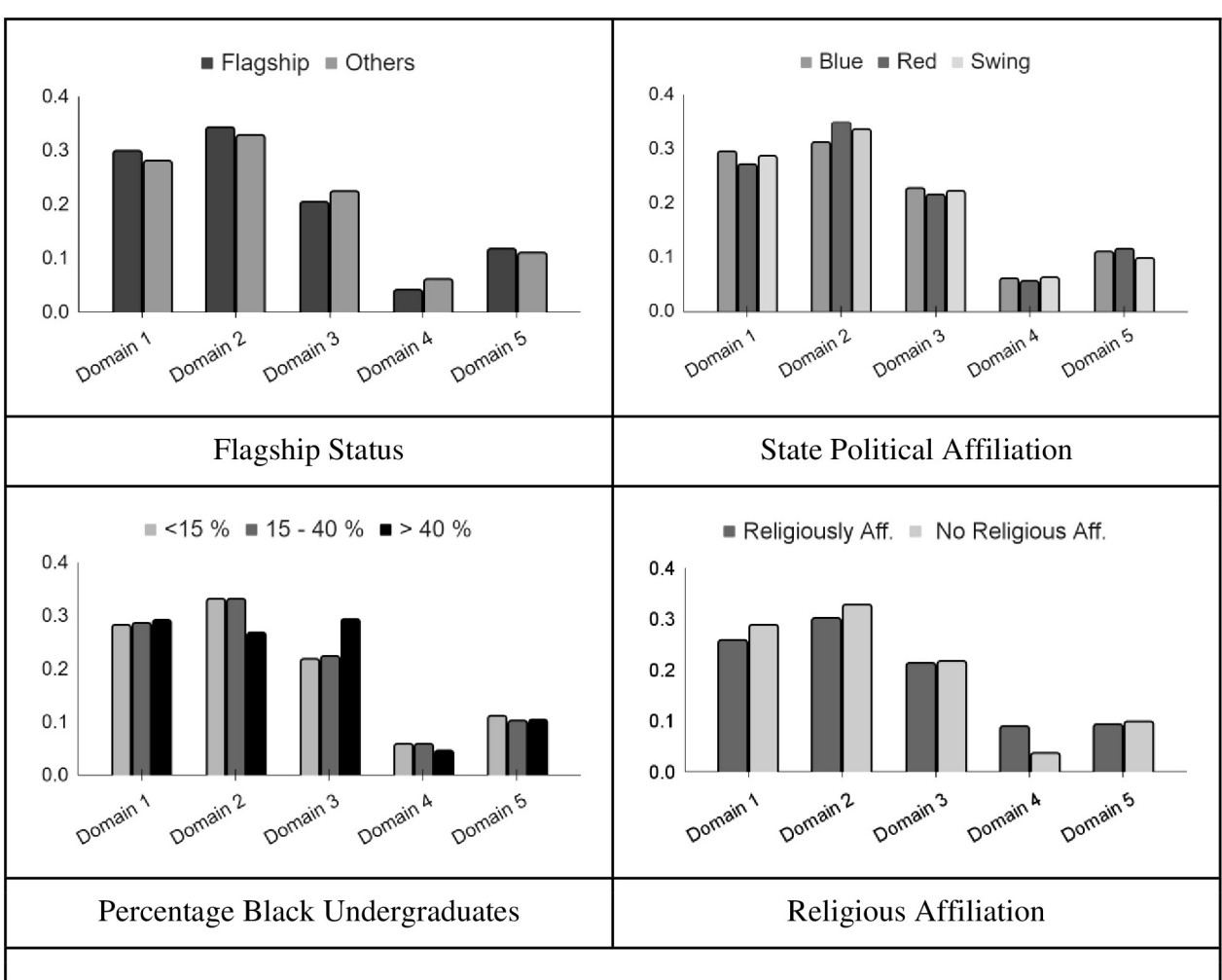

**Fig 1. Histograms of topic domain distributions by IHE attributes.** Legends for each panel are unique and shown above each histogram. Substantive descriptions of the domains for all panels are noted at the bottom of the figure.

The bar graphs in Fig 1 show that flagship schools are somewhat more likely to emphasize racial violence themes ('Racism and Racial Violence' and 'Institutional Reckoning and Response') and less likely to draw on religious themes captured in 'Christian and Humanist Values'. Yet, Table 4 shows that, having accounted for other variables, flagship status has no bearing on connectedness in the network. HBCUs, in contrast, are more connected overall, but are not especially likely to be connected to each other. The implication is that HBCUs tend

to share themes with many other IHEs or that schools draw on themes frequently used by HBCUs. A few rows down, however, we find that IHEs with a higher percentage of Black undergraduates are also not more likely to be connected to others. At the same time, the table shows evidence in favor of homophily by Black student percentage, indicative of clustering of themes based on this variable. Thus, while HBCUs are not systematically more likely to be directly connected to one another, IHEs with higher percentages of Black students do tend to draw on similar themes. Fig 1 clarifies that sharedness of themes based on this variable is likely attributable to higher investedness in topics related to 'Rhetoric on Race as a Historical Social Problem.' Moreover, IHEs with greater than forty percent Black students are considerably less likely to draw on themes related to diversity. Note that HBCUs are all included in this category. As such, IHEs with a high concentration of Black students are more likely to draw on topics linked to RFT and CRT, maintaining that race is a social construct and that racial categories are the basis of differential treatment within political, social, and economic systems.

In the case of ranks, lower numerical values are indicative of higher prestige (i.e., a rank of ten is higher status than a rank of thirty). Thus, the negative estimate for 'Rankings Activity' in Table 4 shows that prestigious IHEs are likely to be more connected in the network. This is likely because elite IHEs often serve as role models for other institutions looking for guidance. High-ranked IHEs are also more likely to be connected to each other via shared themes. Topic distribution by rankings (not shown) demonstrates that, much like HBCUs, elite IHEs are more likely to be invested in topics related to racial violence and the legacy of racism in the United States and are less focused on institutional response and humanist sentiment. These findings further support a shift from colorblind and diversity-based rhetoric in IHEs as demonstrated in prior literature, and especially with regard to HBCUs and elite institutions.

Interestingly, the time a statement was released has no bearing on connectedness. The implication is that having accounted for sharing and popularity based on other attributes, statements released earlier are not more likely to have been used as templates for future statements. This offers some weak evidence against the general diffusion of themes over time, though more rigorous analysis is needed for confirmation. Keep in mind, however, that statements released by HBCUs and elite schools do tend to be more popular, indicative of influence from those institutions to others. We find a similar trend for the percentage of the undergraduate population that is female: IHEs with more female undergraduates have higher connectedness in the network. Compositional analysis (not shown) suggests that IHEs with high concentrations of female students are more invested in Domains 1 ('Racism and Racial Violence/Injustice') and 3 ('Rhetoric on Race as a Historical Social Problem'), and less in Domain 2 ('Institutional Reckoning and Response'). There is also evidence in support of 'heterophily' based on this variable: IHEs with discrepant proportions of female undergraduates are more likely to be interconnected via shared themes suggesting that IHEs with smaller proportions of female undergraduates are likely to turn to such schools, that are more invested in critical approaches to racism, for cues on how to frame their own statements.

The penultimate estimate in Table 4 shows that IHEs do not tend to be linked via common themes if they are co-located in the same geographic region. Although seemingly surprising, this finding makes sense in light of the final estimate, which shows a strong tendency towards homophily based on political affiliation. Fig 1 clarifies that IHEs located in red states are significantly less likely than those in blue states to draw on topics related to racial violence, 'Racism and Racial Violence/Injustice' and 'Rhetoric on Race as a Historical Social Problem,' but more likely to focus on institutional responses captured in Domains 2 ('Institutional Reckoning and Response') and 5 ('COVID-19'). The final two estimates taken together suggest that having accounted for being located in states that voted the same way, there is no additional tendency for IHEs located in the same geographic region of the country (Northeast, South, Midwest,

West, and Pacific) to also be interconnected via shared themes. The negative estimate for regional homophily is likely because the geographic regions we used in the model are large classifications composed of politically divergent states. For example, while most states located in the Northeast voted Democrat in the two most recent presidential elections, Pennsylvania, also classified as located in the Northeast, is a swing state. To the extent schools in Pennsylvania are more likely to share themes with other swing states (and possible red states), there is likely to be relatively lower intra-region connectivity. Moreover, blue states tend to be geographically distant located along both coasts and the middle of the country. Accordingly, a tendency to share themes with other IHEs in blue states should produce connectivity spanning regions, contributing to the negative estimate for intra-region connectivity. Overall, these findings suggest that the political leanings of states in which IHEs are located trump potential tendencies for similarities based on regional proximity.

Homophily based on political affiliation is also evident from Fig 2, which shows the one-mode projection of the complete network. The visualization algorithm places sets of nodes closer together if they are more densely connected. IHEs are colored by the political affiliation of the state in which institutions are located. Red nodes depict IHEs located in states that voted for the Republican candidate in the 2016 and 2020 U.S. Presidential election. Blue nodes, likewise, represent IHEs that voted for the Democratic candidate. Finally, purple nodes represent IHEs located in states that voted differently across the two elections. While several schools are rendered isolate because they do not share at least one topic in common with other IHEs (based on the 0.1111 cutoff described in the Methods section), a large section of the network remains densely interconnected. Consistent with the ERGM estimate, the network shows several regions of clustering based on color, illustrated in encircled portions. The implication is that IHEs located in states that voted similarly tend to draw on similar themes in statements. Note also that blue nodes tend to be more dominant in the center

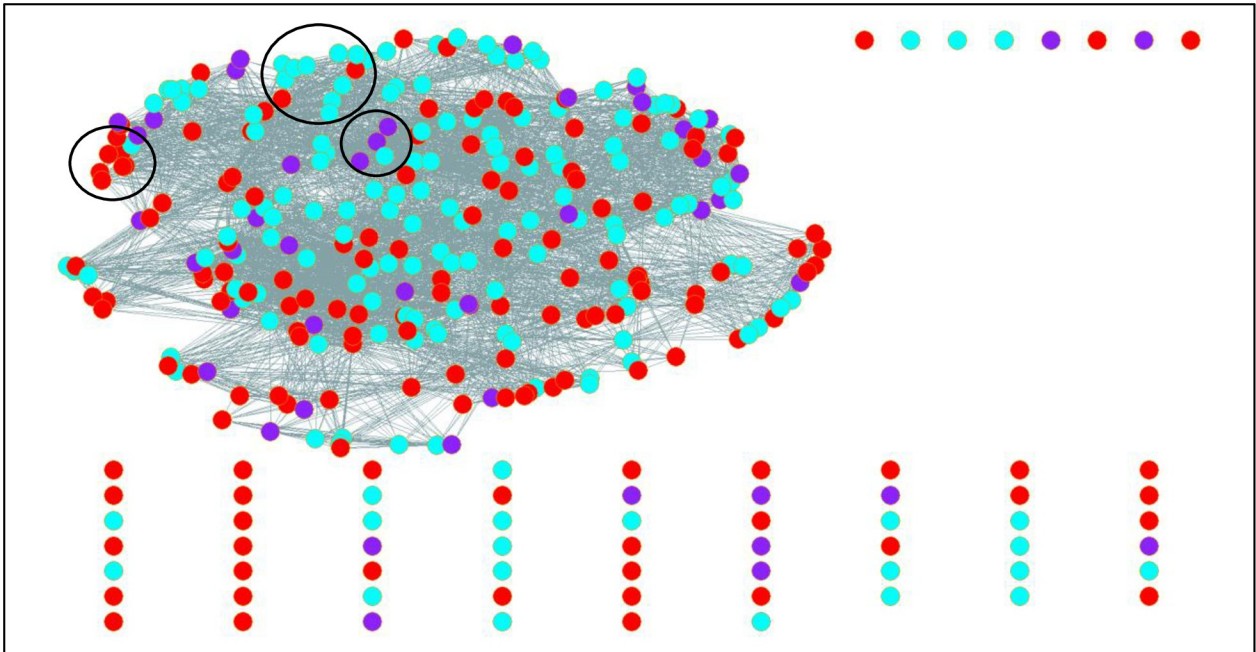

**Fig 2. One-mode projection of the bipartite network.** Node color represents the political affiliation of the state in which the school is located.

of the network suggesting that IHEs located in blue states tend to share more in common with others. Another possible interpretation is that IHEs in blue states used themes that were more central to the corpus.

## Discussion

Prior literature investigating discourse on race by IHEs in the United States has shown a predominance of two models. First, policy and strategy has largely been framed around the theme of diversity and inclusion, what Berrey [26] describes as the diversity orthodoxy. This framing treats race as a cultural identity, emphasizing the benefits of pluralist interactions. As such, it implicitly minimizes the significance of race as a basis for exclusion and disadvantage. Second, in addition to student attitudes being framed by colorblind ideologies, institutional responses to incidents of racism on campus have largely been framed in colorblind terms, positing racial inequalities and racism as matters of personal prejudice. Our findings, based on analyzing statements issued by a large number of colleges and universities in the United States during the Summer of 2020, diverge considerably from this prior literature. As such, our investigation is strongly indicative of a change in the rhetoric used by leadership in American higher education institutions to discuss race and racism.

Most significantly, we find that a large fraction of the corpus is dedicated to explicit discussion of systemic racism as well as racial inequality and injustice. Statements also emphasize that the enduring historical legacy of racial violence against people of color continues to be a reality in the contemporary United States. This is evident in sections of statements that discuss racial disparities and inequities associated with the COVID-19 pandemic, race-based discrimination within workplaces, and police violence targeting Black and other people of color disproportionately. Importantly, in contrast to findings from prior literature, which demonstrates that IHE leadership has tended to emphasize individual prejudice to account for past occurrences of racism on campus, when incidents of discrimination and racial violence are mentioned in statements from the Summer of 2020, they are typically framed as part of a larger pattern of racism rather than the consequence of individual biases. Statements also frequently draw a direct line from the legacy of slavery and segregation in the United States to current incidents of police brutality. In total, fifty percent of the total corpus, encapsulated in Domains 1 ('Racism and Racial Violence/Injustice') and 3 ('Rhetoric on Race as a Historical Social Problem'), is dedicated to this type of discourse. These findings are consistent with what Warikoo and de Novais [5] call the 'power analysis' race frame as well as with core tenets of Critical Race Theory and Racial Formation Theory.

In contrast to these themes and prior literature, we find relatively lower prevalence of colorblind ideologies in the data. Sections of statements captured in Domain 4, 'Christian and Humanist Values,' come closest to mirroring this paradigm. By positing that 'all humans are created equal,' this discourse effectively disregards the role of race as a basis for discrimination and violence in the United States. But here, too, some statements make tacit references to racism, condemning it as amoral and ungodly. Moreover, as noted in the Results section, the standard deviation of the proportion of statements dedicated to this topic is higher than any other in the corpus. The implication is that there is considerable disparity between IHEs in invoking rhetoric associated with this topic. In contrast, the standard deviation of topic investedness is lowest for the first two topics ('Racial Injustice' and 'Racial Violence and Black Lives Matter') in Domain 1 ('Racism and Racial Violence/Injustice'), suggesting that there is greater consistency in the use of these themes across universities. The low prevalence of colorblind ideology in these statements, too, stands in contrast to findings from prior literature, which shows the dominance of this theme in conversations on race and racism in United States IHEs.

We posit that this change in the rhetoric from the invocation of colorblindness to account for incidents *within* the school context, as shown in prior scholarship, to emphasizing systemic racism as prevalent in the *broader* society, as shown in our analysis, marks a remarkable shift in the rhetoric associated with discourse on race in United States IHEs. The extensiveness of this transformation in rhetoric used by university leadership is suggestive of language surrounding systemic racism being poised to acquire orthodoxy-like status in this field. Similar to the emergence of diversity orthodoxy, changes in the political climate, especially a surge of social activism in the Summer of 2020 when these statements were released, could have played an important role in provoking this shift in rhetoric. The attributional analysis is also indicative of conditions conducive to this outcome. Specifically, elite IHEs, which are more invested in the first ('Racism and Racial Violence/Injustice') and third ('Rhetoric on Race as a Historical Social Problem') domains, have greater centrality in the network. Research in neo-institutionalism demonstrates that organizations face pressures to conform to prevailing models, especially once their elite peers adopt a new way of doing things [56]. The spread of diversity orthodoxy has also been argued to have been subject to similar imitational pressures [26, 57–59]. This raises the possibility that discourse reflective of tenets from Critical Race Theory and Racial Formation Theory will become more codified over time as IHEs continue to draw on themes used by their elite peers in such statements. There are, however, three caveats to this potential shifting landscape.

First, despite the centrality of elite IHEs, the analysis reveals unevenness in themes invoked by IHEs based on their attributes. IHEs are more likely to draw on the same topics if they are co-located in states that tend to vote for the same political party in presidential elections. Similar patterns of fragmentation are evident based on IHE religious affiliation. Significantly, IHEs located in states that tend to vote Republican and those that have a religious affiliation are relatively less invested in themes focused on systemic racism and more in diversity orthodoxy. In contrast, IHEs with high percentages of Black students are more likely to highlight the historical legacy of racism and to frame racism as a systemic social problem. The implication is that invocation of discourse that draws on RFT and CRT–an already deeply polarized landscape [60]–may be uneven, both reflecting and deepening existing political divisions.

Second, language steeped in diversity, equity, and inclusion, captured in Domain 2 ('Institutional Reckoning and Response'), continues to be a significant part of the rhetoric, occupying a third of the entire corpus. Akin to Domain 4 ('Christian and Humanist Values'), focused on humanist values, some IHEs are deeply invested in topics covered under this domain. Nearly seventy percent of Stevens Institute of Technology and more than half of University of North Dakota, Drake University, Indiana State University, and University of California at Riverside statements are dedicated to discussion of topics within 'Institutional Reckoning and Response.'. At the lower end, about fifteen percent of statements issued by Arizona State University at Tempe and Worcester Polytechnic Institute draw on diversity-related themes. Thus, it is not the case that diversity orthodoxy has been supplanted by rhetoric that draws on CRT and RFT. Indeed, all IHEs in our dataset were considerably invested in themes focused on diversity, equity, and inclusion.

It is also worth noting that solutions offered by IHEs differed in scale depending on topics invoked in the corpus. On the one hand, diversity-based race frames offered campus-bound solutions to racial injustice and violence such as openness to dialogue and commitment to safe spaces. In contrast, when focusing on broader society, university statements emphasized solutions in their own wheelhouse: leveraging research and education to examine the causes and consequences of systemic racism, and also to inform action in confronting racism as a national social problem. This two-pronged investment in solutions consistent with diversity orthodoxy as well as CRT and RFT associated focus on systemic racism suggests that the former may

coexist alongside an emergent orthodoxy focused on the latter. Further analysis on the implementation of solutions proffered as well as analysis of rhetoric used on an ongoing basis is necessary to adjudicate the balance between the two epistemologies.

The third and final caveat is that there are limits to acknowledgement of systemic racism in these statements. Despite frequent mentions of George Floyd, Eric Garner, Breonna Taylor, Ahmaud Arbery, and others, rhetoric framing violence against Black people explicitly as perpetrated by the police was especially rare. This finding is consistent with prior work analyzing smaller samples of schools. The term 'police brutality,' arguably the most clearly formulated description of police violence (as opposed to more vague and coded rhetoric such as 'excessive use of force') appears in only two percent of all statements we analyzed. Other than IHEs with a high percentage of Black undergraduates (where it was ten percent), about five percent of the total corpus was invested in the topic we call 'Racial Police Brutality.' Even within this topic, it was more common to refer to police involvement obliquely rather than directly. Likewise, while statements explicitly discussed instances of racism and violence across the nation, it was rare for the issuing IHE to name itself or academia, more broadly, as also implicated in systemic racism.

When compared with findings from prior literature, these results are consistent with a second striking shift in the rhetoric on racism in United States higher educational institutions. Earlier research shows that statements focused on incidents within school settings frequently addressed the 'racist' but not the broader context of racism. Our analysis reveals that in releasing statements focused on addressing societal social problems in the Summer of 2020, IHE leadership frequently drew on the opposite framing: acknowledging the diffuse systemic racism evident in society's myriad structures, but largely refraining from explicitly naming any perpetrators.

To conclude, we agree with political commentators that the social unrest in the Summer of 2020 marked a historic moment in the United States. The institutionalization of diversity discourse and colorblind racism within higher education over the last two decades of the twentieth century created conditions for university administrators to release statements in response to these events. We treat these responses as data indicative of IHE positions on these incidents as well as racism in society, more generally. Our results, based on a rigorous analysis of 356 statements issued by IHEs in the United States, show the prevalence of several themes as well as convergence and fragmentation in the discourse depending on IHE attributes. When we compare our results to prior literature, we find evidence supportive of shifts in both the content and form of discourse on race and racism in higher educational institutions. With respect to *content*, our results show a suppression of talk focused on colorblind racism. And, while rhetoric focused on diversity remains high, we also find considerable investedness in something that was previously largely missing—tenets of CRT, especially in IHEs acknowledgement of the structural nature of racism in the United States. This finding suggests that language surrounding CRT is well-positioned to becoming institutionalized in U.S. colleges and universities. On *forms* of talk, we find divergence from prior literature which shows that, when dealing with on-campus incidents, university leadership tended to emphasize individual 'racists' but shied away from drawing attention to systems implicated in acts of racism. In contrast, our corpus of IHE responses from the Summer of 2020 reveals little mention of the perpetrators of racial violence but contains considerable affirmation of the diffuse and historical legacy of racism in the United States. These shifts, we believe, are largely possible because IHEs in the United States remain engaged in critical race discourse, regularly producing research that demonstrates the systemic nature of racism. Yet, it remains to be seen if Critical Race Theory continues to feature prominently in IHE rhetoric and if commitments to address systemic racism made in these statements come to bear in the years to come.

## Author Contributions

**Conceptualization:** Noor Toraif, Neha Gondal.

**Data curation:** Noor Toraif, Alison Frisella.

**Formal analysis:** Noor Toraif, Neha Gondal, Pujan Paudel, Alison Frisella.

**Methodology:** Neha Gondal.

**Project administration:** Neha Gondal.

**Writing – original draft:** Noor Toraif, Neha Gondal.

**Writing – review & editing:** Noor Toraif, Neha Gondal.

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
