## [Decision Letter · Decision Letter 0]

6 Sep 2022

PONE-D-22-17413

“We are saddened by the tragic events of the last…”: A topic modeling analysis of responses by Institutions of Higher Education in the United States to the murder of George Floyd

PLOS ONE

Dear Dr. Gondal,

Thank you for submitting your manuscript to PLOS ONE. After careful consideration, we feel that it has merit but does not fully meet PLOS ONE’s publication criteria as it currently stands. Therefore, we invite you to submit a revised version of the manuscript that addresses the points raised during the review process.

We look forward to receiving your revised manuscript.

Kind regards,

Hyejin Youn

Academic Editor

PLOS ONE

Journal Requirements:

Additional Editor Comments:

Two reviews suggested the decision drastically differently, but when you read Reviewer 1's point-by-point issues and suggestions are broadly consistent with Reviewer 2. Addressing these issues will greatly improve the current manuscript. 

Reviewers' comments:

Reviewer's Responses to Questions

**Comments to the Author**

1. Is the manuscript technically sound, and do the data support the conclusions?

Reviewer #1: Yes

Reviewer #2: No

2. Has the statistical analysis been performed appropriately and rigorously? 

Reviewer #1: Yes

Reviewer #2: No

3. Have the authors made all data underlying the findings in their manuscript fully available?

Reviewer #1: Yes

Reviewer #2: Yes

4. Is the manuscript presented in an intelligible fashion and written in standard English?

Reviewer #1: Yes

Reviewer #2: Yes

5. Review Comments to the Author

Reviewer #1: I advise an accept given that my questions and comments are answered. Authors exploited a novel idea of analyzing statements on a salient issue by educational institutions with a machine-learning approach. Authors analyze statements released on a white police officer's murder of George Floyd. The statements were collected from 356 institutions of higher education and investigated using topic modeling and ERGM. From the corpus, authors identify 18 topics in 5 domains for the discourses on race.

Please see the attachment for comments.

Reviewer #2: Dear editors,

Thanks for the opportunity to review the manuscript using computational methods to investigate how the US higher education institutions responded to George Floyd’s murder. The topic is socially important, but I found many issues with the manuscript’s organization, argument, and methods.

Below is the list of my concerns.

1. The title is too long. Unless the method is novel or the authors developed the method, I don’t recommend including the method name in the subtitle. Instead, the title should focus on the main question, evidence, and implications.

2. The same problem exists for the abstract. I recommend avoiding adjectives, such as “fundamentally,” unless they are essential for the argument.

3. The sampling frame is vague. The authors stated that N = 356 are most PhD-granting institutions in the US. How most are they? Put differently, are their representative of the population (PhD-granting institutions)? Also, why only focus on Ph.D. granting institutions? This decision rules out more teaching-oriented institutions such as SLACs and community colleges in the population. How was such a case selection strategy justified on what grounds?

4. The description of the colorblindness approach is vague. What does it mean that colorblindness treats racism “as a historical phenomenon.” Further elaboration on it is required. What’s the relationship between colorblindness and institutional responses? Many concepts are loosely defined and connected.

5. The description of the diversity theme is inadequate. Colorblindness is an ideology, so the diversity theme conceptualization should also be grounded in ideology. Alternatives: Multiculturalism? Racial equity and equality? Calling it diversity is too vague.

6. Going back to point 3, the authors' conceptualization of the population is the Institute of Higher Education (p.4). There’s a leap from this population to the data used in the manuscript. A justification is needed. Also, it's unclear what the authors mean by the sample was selected based on the ranks produced by the US News and World Report in 2021 (also, why this index is relevant here?).

7. The data interpretation is not well connected to the theory. The theory assumes that two prevalent ideological themes influenced how the US colleges issued statements regarding the murder of George Floyd. How do we know that’s the case? Perhaps, the underlying cause is the institution's strategic decisions based on their constituencies. The authors did this type of reasoning when they mentioned how the prevalence of mentioning perpetrators varies by the school’s locations and their partisan characteristics (p.11, p.23). In short, many interpretations in the manuscript relied on post-hoc theorization. On the minor related point, it’s surprising that the manuscript didn’t mention “Black Lives Matter” only twice and did not delve into it! Why there's no Black Lives Matter protest variable included in the estimation model?

6. PLOS authors have the option to publish the peer review history of their article (what does this mean?). If published, this will include your full peer review and any attached files.

Reviewer #1: No

Reviewer #2: No

---

## [Author Response · Author response to Decision Letter 0]

26 Sep 2022

We would like to thank the reviewers for their detailed comments and suggestions. We believe we have benefitted from a diligent re-working of the paper. A significant chunk of modifications occurs in the ‘Introduction’ section. Here, we have enriched our discussion of ‘diversity,’ describing it as an orthodoxy in Institutions of Higher Education (IHEs). We have also expanded on our discussion of ‘systemic’ racism by drawing on two major theoretical frameworks – Racial Formation Theory (RFT) and Critical Race Theory (CRT). We subsequently invoke these theories in the discussion of our results with the goal of greater coherence between the Introduction and Results sections. Finally, we bring the threads together in our ‘Discussion’ section, connecting our findings to colorblindness, diversity orthodoxy, RFT, and CRT. We have also provided more details about our choice of data, in response to suggestions and questions from both reviewers. Details of the changes we have made are below.

Reviewer 1

1. Page 9, Lines 165-167: Is there a case of releasing multiple statements from the same institution? If so, how did you deal with such cases? 

Yes, there were few instances of multiple statements released by the same institution. If an institution released multiple statements, only the first released statement was included in our sample. We had included this information in the first draft on pages 8-9, Lines 158-159. That information has been retained in the current version of the manuscript on Page 10, Lines 229-230.

2. Page 9, Line 172: Please include a descriptive statistics of documents such as document length released by each university. It seems that Table 1 only has the summary statistics of the universities.

We thank the reviewer for drawing our attention to this blind spot. We have included data on statement length (in words and sentences) as well as time of release in Table 1.

3. Page 10, Line 180: When was the dataset of College Factual accessed? It would be helpful to include the year that the College Factual dataset is released.

We accessed data from College Factual for all variables in the month of August, 2021. This information is included in the revised Manuscript on Page 13, Line 263. 

4. Page 11, Lines 212-213: Please include your reasoning of using the highest number – 42 days – not the mean value (or some other days) for the statements with missing time stamps.

We had originally fit the model using mean and median days for the missing data as well as the maximum value. We failed to include that information in the original manuscript. Much like the maximum value, neither the mean nor the median resulted in statistically significant estimates. Indeed, the estimates were nearly indistinguishable regardless of the values we used in the model. We have included the following information in the revised manuscript on Page 14, Lines 291-293:

“We also tried the mean as well as median number of days in place of the maximum. The results of the estimation remained unchanged regardless of the value used for the missing data.”

5. Page 12, Lines 223-228: Do the documents meet assumptions for Latent Dirichlet Allocation?

Yes, the documents meet the assumptions for use of LDA on the corpus. We have added some information elaborating on this on Page 15, Lines 308-311.

6. Page 12, Lines 231-232: Please try additional ways of checking the optimal number of topics systematically such as log-likelihood or perplexity.

In addition to the coherence score, we also tested the log-likelihood for our models. The log-likelihood scores largely support our decision based on the coherence score and substantive reasoning. We have added the following text to the revised manuscript on Page 15, Lines 324-324:

“We also ran the log-likelihood associated with each model. Among models with at least fourteen topics, our preferred solution with eighteen topics had the lowest log likelihood. Only models with ten or twelve topics had slightly lower scores, but, as stated above, we found those solutions to be substantively inadequate.”

7. Page 13, Line 239: Add examples of tokens (e.g., tokenized verbs).

We have added examples of tokens on Page 16, Lines 328-333.

8. Page 13, Lines 258-260: Include network visualization of the bipartite graph – either the full set of topics and statements at a glance or a subset of them.

We have included a network visualization of the one-mode projection of the complete network based on the strategy described in the paper. The original two-mode network of statements and topics was not visually appealing or informative. Instead, we found a visualization based on political affiliation of states in which schools are located to be more interesting. This is included as Figure 2 and described as follows on Page 35, Lines 656-670:

“Homophily based on political affiliation is also evident from Figure 2, which shows the one-mode projection of the complete network. The visualization algorithm places sets of nodes closer together if they are more densely connected. Schools are colored by the political affiliation of the state in which institutions are located. Red nodes depict schools located in states that voted for the Republican candidate in the 2016 and 2020 U.S. Presidential election. Blue nodes, likewise, represent schools that voted for the Democratic candidate. Finally, purple nodes represent schools located in states that voted differently across the two elections. While several schools are rendered isolate because they do not share at least one topic in common (based on the 0.1111 cutoff described in the Methods section) with other schools, a large section of the network remains densely interconnected. Consistent with the ERGM estimate, the network shows several regions of clustering based on color, illustrated in encircled portions. The implication is that schools located in states that voted similarly tend to draw on similar themes in statements. Note also that blue nodes tend to be more dominant in the center of the network suggesting that schools located in ‘blue’ states tend to share more in common with others. Another potential interpretation is that blue states used themes that were more central to the corpus.”

9. Pages 16-17, Lines 328-329: Include additional explanations on how you define the ‘race-centric’ issues and group (i.e., domain) of topics.

Thank you again for bringing this and the next point to our attention. We define degree of race-centricness based on the number of tokens in top fifteen keyworks that spoke directly to what we considered to be issues focused on race. We add the following text to elaborate on the concept on Page 20, Lines 421-427:

“These are issues that we considered to be explicitly focused on race in the United States. Topics with a high concentration of such issues contained several explicitly race-related tokens in their top keywords such as ‘black,’ ‘discrimination,’ ‘systemic racism,’ and ‘equity’. Topics with low prevalence of such issues contained almost no such tokens. The top-fifteen tokens in the second topic in Domain 5, for example, contain almost no tokens that we consider to be expressly race-related issues.”

10. Pages 23, Lines 370. What do you mean by “invested" in a topic? Add a line or two for explanation.

We added the following text to elaborate on topic investedness. This appears on Page 16, Lines 347-351:

“An edge in this network denotes the proportion of the statement being composed of a given topic. We refer to this also as the degree of a statement’s ‘investedness’ in a topic. A university or college is more invested in a topic, for example, when a higher proportion of its statement is devoted to that topic.”

11. Page 29, Lines 493-494. The region homophily has a negative value in Table 3. On the other hand (Page 30, Lines 527-529), schools tend to have the same topics if co-located in some states. Is it for `red' and `blue' but not `swing' states? Please include supporting evidence.

This is an important point that we failed to adequately explain in the first version of the manuscript. The region variable is a coarse-grained division with five categories: Northeast, South, Midwest, West, and Pacific. Political affiliation, on the other hand, links two schools if they voted the same way in the two most recent presidential elections. As we explain in the manuscript, there are two likely reasons for negative regional homophily alongside positive political homophily. First, regions are composed of politically divergent states such as New York and Pennsylvania. Second, politically convergent Democrat-voting states are located at great distances – along the two coasts and the middle of the country. As such, convergence of themes based on political similarity generates a negative estimate for regional homophily. This explanation can be seen on Page 34, Lines 635-654.

Reviewer 2

1. The title is too long. Unless the method is novel or the authors developed the method, I don’t recommend including the method name in the subtitle. Instead, the title should focus on the main question, evidence, and implications.

We have amended the title to be more substantively focused.

2. The same problem exists for the abstract. I recommend avoiding adjectives, such as “fundamentally,” unless they are essential for the argument.

We have also amended the abstract to focus on the primary question and findings.

3. The sampling frame is vague. The authors stated that N = 356 are most PhD-granting institutions in the US. How most are they? Put differently, are their representative of the population (PhD-granting institutions)? Also, why only focus on Ph.D. granting institutions? This decision rules out more teaching-oriented institutions such as SLACs and community colleges in the population. How was such a case selection strategy justified on what grounds?

We thank the reviewer for drawing our attention to this error. An earlier draft of the paper had focused only on PhD granting institutions, and that language slipped into the previously submitted version of the manuscript in error. We have expanded considerably on our choice of data in this revised version of the manuscript. Specifically, we focused on all colleges and universities included in the ‘National Universities’ rankings produced by U.S. News and World Report (USNWR) in 2021 (N=388). We explain in some detail why USNWR is a reasonable choice for delineating a data frame. Most significantly, we are interested in the effect of prestige and rankings on theme choice and connectedness of IHEs by theme. This would not be possible if we chose schools spanning lists. Our revised text, reproduced below, appears on Page 9, Lines 202-222:

“We focused on all IHEs included in the ‘National Universities’ rankings produced by U.S. News and World Report (USNWR) in 2021 (N=388). USNWR started publishing evaluations of colleges and universities in the early eighties. Although rankings produced by the organization have received some criticism, USNWR has come to garner tremendous legitimacy as an evaluator of universities and colleges in the U.S. [44, 45]. Their evaluations are based on a variety of indicators of academic quality such as graduation rates, faculty and student resources, and admissions selectivity. UNSWR draws on the Carnegie Classification for categorizing IHEs, a widely accepted standard in the U.S., to produce several distinct types of rankings. We draw on schools ranked in the ‘National University’ category, which includes colleges and universities that offer a range of undergraduate degrees, master’s programs, as well as doctoral degrees. These schools are also at the forefront of academic research. As such, we exclude institutions focused primarily on undergraduate education such as liberal arts colleges, regional schools, and community colleges. Our primary reason for focusing on graduate-degree granting institutions is that much prior research on institutional responses has investigated such schools. Second, our goal of investigating the effect of rankings on shared themes is only feasible if institutions are ranked on the same evaluation system. Undergraduate schools, for example, are evaluated using different metrics owing to their distinctive organizational structure. Accordingly, it would be hard to reconcile and appropriately compare schools ranked across lists (such as Williams College, ranked highly in liberal arts schools, and Princeton University, ranked highly among National Universities).”

4. The description of the colorblindness approach is vague. What does it mean that colorblindness treats racism “as a historical phenomenon.” Further elaboration on it is required. What’s the relationship between colorblindness and institutional responses? Many concepts are loosely defined and connected.

We have clarified that by ‘historical phenomenon,’ we mean something that largely existed in the past. We have replaced that language in the abstract to offer more clarity. 

We have also elaborated on the relationship between colorblind ideology and institutional responses in the second paragraph on the ‘Introduction’ section. Here, we describe the results from several studies that show that college students generally tend to adhere to colorblind ideologies while in college. We also elaborate on two studies that demonstrate the use of colorblind frames in institutional responses to racist incidents on campus. The amended text appears on Page 2, Lines 41-57.

5. The description of the diversity theme is inadequate. Colorblindness is an ideology, so the diversity theme conceptualization should also be grounded in ideology. Alternatives: Multiculturalism? Racial equity and equality? Calling it diversity is too vague.

Thank you for raising this. Responding to this suggestion produced a significant addition in our manuscript. Drawing on the work of Ellen Berrey, we now elaborate on the ‘diversity’ paradigm, describing it as an orthodoxy, and accounting for its emergence and institutionalization in Institutions of Higher Education. The following amended text appears on Page 3, lines 67-85.

“As distinct from an ideology, which provides a template for the organization of the world [1], an orthodoxy constitutes a set of widely shared ideas, beliefs, and practices that guide institutional discourse as well as policy, strategy, and action. 

Berrey [24] argues that, over the last two decades of the twentieth century, “diversity” became a keyword in United States Institutions of Higher Educations’ (IHEs) policies and programs surrounding race. This shift occurred, in part, due to organizational pressures in a changing political, demographic, and legal climate. An early impetus can be traced to a minority opinion issued in a significant legal case challenging affirmative action admissions policies in the late seventies. This case laid the groundwork for using diversity as a rationale for race-conscious admissions and subsequent contentious lawsuits, both challenging and supporting such policies, helped codify language surrounding diversity. Thereafter, shifts in demographics of the college going population - a rise in immigrants, foreign students, women, and people of color generated greater need for strategy and rhetoric to manage heterogeneous student populations. These strategies diffused rapidly across academic institutions becoming normative and exerting pressure on others to signal their own commitment to inclusiveness. Indeed, diversity acquired so much popularity over time that it came to replace the formerly reigning buzzword, ‘multiculturalism,’ in higher education rhetoric [22].” 

6. Going back to point 3, the authors' conceptualization of the population is the Institute of Higher Education (p.4). There’s a leap from this population to the data used in the manuscript. A justification is needed. Also, it's unclear what the authors mean by the sample was selected based on the ranks produced by the US News and World Report in 2021 (also, why this index is relevant here?).

Please see our response to point 3 above where we clarify how our data are linked to U.S. News and World Report Rankings. We believe that our dataset, containing a mix of colleges and universities that offer a range of undergraduate degrees, master’s programs, as well as doctoral degrees, is a reasonable, though not perfect, representation of Institutions of Higher Education in the United States. Accordingly, we believe that the IHE phrasing is appropriate. 

7. The data interpretation is not well connected to the theory. The theory assumes that two prevalent ideological themes influenced how the US colleges issued statements regarding the murder of George Floyd. How do we know that’s the case? Perhaps, the underlying cause is the institution's strategic decisions based on their constituencies. The authors did this type of reasoning when they mentioned how the prevalence of mentioning perpetrators varies by the school’s locations and their partisan characteristics (p.11, p.23). In short, many interpretations in the manuscript relied on post-hoc theorization. On the minor related point, it’s surprising that the manuscript didn’t mention “Black Lives Matter” only twice and did not delve into it! Why there's no Black Lives Matter protest variable included in the estimation model?

This comment contains several points. We believe we have addressed all of them in the revised version of the manuscript. 

First, we did not make adequate connections between our results and theory in the previous version of the manuscript. We have aimed to redress this issue by systematically linking our findings to colorblind ideology and diversity orthodoxy in this revised version. 

Second, we have added several paragraphs to the ‘Introduction’ section elaborating on what Warikoo and de Novais (2015) call the ‘power analysis’ frame. Specifically, we argue that this framing draws on two dominant theories of race in the social sciences - Racial Formation Theory (RFT) and Critical Race Theory (CRT). We elaborate on the core tenets of each of these theories, situating them in opposition to colorblind ideology and diversity orthodoxy. Most significantly, both theories emphasize the pervasive and enduring nature of racism in the United States. In addition to colorblindness and diversity, we also draw on CRT and RFT to make sense of our findings in the ‘Results’ as well as the ‘Discussion’ sections. 

Third, there is nothing in our analysis that suggests that IHEs are not acting strategically. To the contrary, drawing on existing tropes focused on diversity, equity, and inclusion, as well as newer ones focused on systemic racism, may well be purposeful action aimed at appearing to be committed to inclusiveness. 

Finally, we considered how we might incorporate ‘Black Lives Matter’ into our ERGM model. We looked for data on protests in cities and towns, but found that (1) the available data were not good enough quality, failing to account for all occurrences of protest, for example, and (2) when we looked to create our own data, we found that protests occurred in most towns and cities, and all states, making it very difficult to conceptualize as a variable. We would like to emphasize that one of the topics in our topic modeling analysis is labeled ‘Black Lives Matter.’ We provide details about that topic, including the number of times the trigram ‘Black Lives Matter’ appears in the corpus, on Page 25, Lines 440-446.

---

## [Decision Letter · Decision Letter 1]

24 Jan 2023

PONE-D-22-17413R1From Colorblind to Systemic Racism: Emergence of a Rhetorical Shift in Higher Education Discourse in Response to the Murder of George FloydPLOS ONE

Dear Dr. Gondal,

Thank you for submitting your manuscript to PLOS ONE. After careful consideration, we feel that it has merit but does not fully meet PLOS ONE’s publication criteria as it currently stands. Therefore, we invite you to submit a revised version of the manuscript that addresses the points raised during the review process.I would recommend the authors to address the remarks from all the reviewers especially those flagged by Reviewer 4 about the causality effect of the case of George Floyd.

We look forward to receiving your revised manuscript.

Kind regards,

Maurizio Fiaschetti

Academic Editor

PLOS ONE

Journal Requirements:

Reviewers' comments:

Reviewer's Responses to Questions

**Comments to the Author**

1. If the authors have adequately addressed your comments raised in a previous round of review and you feel that this manuscript is now acceptable for publication, you may indicate that here to bypass the “Comments to the Author” section, enter your conflict of interest statement in the “Confidential to Editor” section, and submit your "Accept" recommendation.

Reviewer #1: All comments have been addressed

Reviewer #3: (No Response)

Reviewer #4: All comments have been addressed

2. Is the manuscript technically sound, and do the data support the conclusions?

Reviewer #1: Yes

Reviewer #3: Yes

Reviewer #4: Partly

3. Has the statistical analysis been performed appropriately and rigorously? 

Reviewer #1: Yes

Reviewer #3: Yes

Reviewer #4: Yes

4. Have the authors made all data underlying the findings in their manuscript fully available?

Reviewer #1: Yes

Reviewer #3: Yes

Reviewer #4: Yes

5. Is the manuscript presented in an intelligible fashion and written in standard English?

Reviewer #1: Yes

Reviewer #3: Yes

Reviewer #4: Yes

6. Review Comments to the Author

Reviewer #1: I advise an accept. Authors provided thoughtful answers to my questions and comments. I am happy that authors significantly improved the paper.

Please see the attachment for minor comments.

Reviewer #3: I can see significant improvement between the two versions. Overall, the methods are well-described and easy to follow. The results are interesting and informative. However, I still have several comments for changes, many of which are minor.

1. I appreciate the inclusion of RFT and CRT. However, I think they are not well integrated into the rest of the paper. I was hoping, when reading the explanation of your LDA results, that you could tie the topics more to the two theories. When reading L132-136, I was wondering how this trend would be reflected in your analysis, but I think the current Results section hasn't elaborated on this.

2. Table 1: Is it meaningful to report the mean or median of school rankings?

I'm also curious about the min and max of many items.

3. L352: et al[.]

4. Not sure if it's a formatting error: Figure 1 and Figure 2's captions and the figures are on separate pages.

5. L585, I wonder if it's possible to label subfigures?

6. L592: May want to mention Figure 1 just for clarity?

7. L593: Table 3?

8. L640: "in concern", not concert?

9. I think discussions starting from L702 should be first presented in the Results section.

10. This is more of only a comment: I think the paper ends rather abruptly. I was hoping for at least a summary of findings, if possible, a bit more implications for future research.

Reviewer #4: The article discusses the racial formation theory (RFT) and critical race theory (CRT) through the university's official statements to the case of George Floyd's murder happened in May 2020. Authors use the topic model analysis as well as the ERGM model to understand the general distribution of university's opinion on this issue and which factors make university to have the same ideas.

While authors carefully collected the data, analyzed them, and made conclusions, I found one major mismatch from the authors' arguments and the data and methods. From the title, "From colorblind to systemic racism," the abstract, "our analysis reveals two striking rhetorical shifts on racial discourse," and several places in the manuscript, authors emphasized that this article reveals the transition of discourse. However, what authors analyzed is the case of George Floyd's murder happened in May 2020. Therefore, technically speaking, authors could not reveal the transition in the discourse by looking at the one-time event.

I concurred with authors that this article contributes to the existing literature by revealing the significance of university's view on systematic racism. However, it does not mean the authors' data and methods found that authors found the fundamental shift in the university's view on racism. In order to show that authors should have illustrated how universities reacted to similar events before and after the case of George Floyd's murder and showed the topic distribution change over time.

Given that this path might not be the feasible way to develop the paper further, I recommend to tone down the paper's argument in transition or emergence of universities' view on racism, but focus on its contribution in understanding the case.

7. PLOS authors have the option to publish the peer review history of their article (what does this mean?). If published, this will include your full peer review and any attached files.

Reviewer #1: No

Reviewer #3: No

Reviewer #4: No

---

## [Author Response · Author response to Decision Letter 1]

31 Jan 2023

We would like to thank the reviewers once again for their detailed comments and suggestions. We believe we have benefitted from a diligent re-working of the paper. Details of the changes we have made are below.

Reviewer 1

1. In the abstract: “statements issued in May 2020” is not correct given that the maximum time window for statement issuance does not fall into a single month.

Thank you for pointing this out. We have fixed this error by changing “May 2020” to “the Summer of 2020”. 

2. Page 7, Line 141 and Page 38, Line 723: May 2020 is not Summer. Please be consistent when stating the time.

We have also addressed this error. We are now consistent in our use of “Summer of 2020” whenever we refer to the period when statements were issued. We only use “May 2020” when we speak of the death of George Floyd.

3. Page 10, Line 216: Add references to prior research.

Done.

4. Page 11, Line 244: It is written that the last statement is made 41 days after while it is written as “forty-two” days in Page 14, Line 291.

Thank you for picking up on this. “41” was an error in the previous manuscript. We have fixed this to read “forty-two” now (Line 261).

5. Page 16, Line 351: How do you measure whether a topic composes a large proportion of a statement?

We have offered clarification for how we measure topic composition in statements. Specifically, we arrive at topic investedness from the document-topic probability matrix. In the MS (page 16, Lines 372-374), we clarify that we “deduce this from the document-topic probabilities vector or topic mixture, which shows the estimated proportion of words from a given statement that are generated from all topics.” 

6. Page 35, Line 669: Why are you stating the interpretation as “potential” interpretation?

We used the term to indicate that it was one of two possible explanations. We have replaced the word with "possible" to make it clearer for the reader.

Reviewer 3

1. I appreciate the inclusion of RFT and CRT. However, I think they are not well integrated into the rest of the paper. I was hoping, when reading the explanation of your LDA results, that you could tie the topics more to the two theories. When reading L132-136, I was wondering how this trend would be reflected in your analysis, but I think the current Results section hasn't elaborated on this.

We thank the reviewer for this comment, which has helped improve the MS. Specifically, throughout the results section, we outline how topic domains or individual topics reflect tenets of CRT, RFT, or diversity/colorblind race frames. For example, in describing the first topic domain, we add the following text on page 25, Lines 459-463: 

“In contrast to prior literature which demonstrates the proliferation of colorblindness and diversity orthodoxy in higher education rhetoric, topics in this domain, we find, resonate strongly with the tenets of Critical Race Theory, and especially the notion that systemic racism is deeply embedded within United States’ social, political, and economic institutions.” 

Please refer to the full Results section for further elaboration on CRT and RFT in relation to our findings. 

2. Table 1: Is it meaningful to report the mean or median of school rankings?

This is a good point. We have removed the line for school rankings from Table 1. 

3. I'm also curious about the min and max of many items.

We have added the minimum and maximum of all continuous variables: statement length (Lines 262-263), undergraduate percentage black (Lines 278-279), and undergraduate percentage female (Lines 286-287) in the description of the attribute variables.

4. 3. L352: et al[.]

Done.

5. 4. Not sure if it's a formatting error: Figure 1 and Figure 2's captions and the figures are on separate pages.

The captions appear on the same pages as the figures on our end. Perhaps this is something that is changed in the translation from the word file to the online proof generated for review purposes. We can make sure to fix this issue if the paper is accepted for publication.

6. 5. L585, I wonder if it's possible to label subfigures?

The sub-figures in Figure 1 are labeled on our end (below each bar graph). Again, this might be an issue in the translation process.

7. 6. L592: May want to mention Figure 1 just for clarity?

Done.

8. 7. L593: Table 3?

Our apologies for this error. We have fixed this typo.

9. 8. L640: "in concern", not concert?

We meant “in concert” to indicate “taken together.” We have modified the language to be clearer for readers.

10. 9. I think discussions starting from L702 should be first presented in the Results section.

We have moved several lines from the Discussion section to the Results section (Lines 558-564) to address this concern. We return to this point in the Discussion section (Lines 759-766) but refer the readers to the material presented in the Results section.

11. 10. This is more of only a comment: I think the paper ends rather abruptly. I was hoping for at least a summary of findings, if possible, a bit more implications for future research.

Thank you for this comment. We have added a concluding paragraph to close out the paper.

Reviewer 4

1. While authors carefully collected the data, analyzed them, and made conclusions, I found one major mismatch from the authors' arguments and the data and methods. From the title, "From colorblind to systemic racism," the abstract, "our analysis reveals two striking rhetorical shifts on racial discourse," and several places in the manuscript, authors emphasized that this article reveals the transition of discourse. However, what authors analyzed is the case of George Floyd's murder happened in May 2020. Therefore, technically speaking, authors could not reveal the transition in the discourse by looking at the one-time event. I concurred with authors that this article contributes to the existing literature by revealing the significance of university's view on systematic racism. However, it does not mean the authors' data and methods found that authors found the fundamental shift in the university's view on racism. In order to show that authors should have illustrated how universities reacted to similar events before and after the case of George Floyd's murder and showed the topic distribution change over time. Given that this path might not be the feasible way to develop the paper further, I recommend to tone down the paper's argument in transition or emergence of universities' view on racism, but focus on its contribution in understanding the case.

We would like to thank the reviewer for this suggestion/comment. We wrestled with it for some time to find a way forward. Ultimately, we have made several substantial changes to the MS, which we hope addresses this important concern. 

Succinctly, with these changes, we hope to clarify that the “shift” we emphasize occurs in reference to what is known based on prior investigations of higher educational institutions. This existing body of literature establishes the dominance of colorblind ideology and diversity rhetoric in both student attitudes (Lines 42-50) and institutional responses (Lines 50-57). Moreover, prior literature shows the absence of acknowledgment of systemic racism in IHEs in the United States (Lines 57-63). Our findings show a strong prevalence of this language in statements issued by IHEs in the Summer of 2020. Thus, we make sense of our findings by situating them in reference to this prior work, thereby arriving at the language of a “shifting landscape.” 

This practice of situating work and locating changes on the basis of prior knowledge is quite standard practice in our fields. Nevertheless, we agree that (1) this may not be standard practice in other fields, and, hence, to readers of Plos One, and (2) we did not make it adequately clear in the previous version of the MS that the shift was in reference to prior work. Accordingly, we have made several modifications to the MS:

i. First, we have added language to the final paragraph of the Introduction section to clarify that our analysis contributes to a body of literature that has already established the prevalence of colorblind ideology and diversity rhetoric in higher education in the United States. Specifically, we say (Lines 204-209):

“While we analyze statements released at approximately one point in time, our objective is not limited to analyzing the rhetoric in that set of responses. We also aim to compare the dialogue invoked in the Summer of 2020 to findings from prior literature, which clearly shows the dominance of colorblind ideology and diversity orthodoxy in dealing with issues related to race and racism in U.S. IHEs.”

ii. We have added a modifier (Line 19) to the abstract to clarify that the shift should be interpreted in reference to prior work.

iii. We have elaborated on the second paragraph of the introduction to emphasize the prevalence of colorblindness in IHEs in the U.S (Lines 42-63).

iv. In addition to instances where we had already noted departures from prior literature in the previous round of the MS (e.g., Lines 508-510 and 512-516), we have added materials in several places in the Results section to clarify that our results point to trends that are distinct from prior findings.

v. We have done the same in the Discussion section. Significantly, every time we refer to shift in rhetoric (e.g., Lines 732-734, 742-746,764-766, 768-771, and others) we clarify that the change is only evident when we compare our findings to prior literature, which shows the dominance of colorblind ideology and diversity orthodoxy.

---

## [Editor Report · Decision Letter 2]

23 Mar 2023

PONE-D-22-17413R2From Colorblind to Systemic Racism: Emergence of a Rhetorical Shift in Higher Education Discourse in Response to the Murder of George FloydPLOS ONE

Dear Dr. Gondal,

Thank you for submitting your manuscript to PLOS ONE. After careful consideration, we feel that it has merit but does not fully meet PLOS ONE’s publication criteria as it currently stands. Therefore, we invite you to submit a revised version of the manuscript that addresses the points raised during the review process.

Two Academic Editors have assessed the manuscript, and would like for you to address the below comments:

1. There is some inconsistency in the use of terminology: colorblindness vs colorblind racism vs colorblind ideology vs colorblind racist ideology vs colorblind rhetoric. This is one example. Authors should clarify if these terms are being used interchangeably or if these terms have different meaning. A table of definitions could address this issue and include other terms: diversity, diversity orthodoxy, "interest convergence", "voice-fo-color" thesis, prejudice, and other race-related terms throughout the text.

2. On page 1, there is a vague and convoluted explanation of terminology from a macro perspective (i.e., institutions of higher education) whereas there tend to be more clearly defined from micro or interpersonal perspectives (e.g., colorblindness at the individual level). Given the sensitivity of DEI and race-related terms, vague definitions can do more harm than good.

3. Lines 46-48, seem to reference the use of microaggressions by college students today, but does not directly use the term "microaggression" which is well-defined in the literature and used indirectly to make racially bias insults in professional and institutional settings.

4. Lines 67-68, which states that "...race is viewed as a matter of cultural difference..." needs further explanation and clarification. This should not be difficult given that 5 articles are cited. Perhaps a definition of diversity should begin this discussion. In line 726, race is referred to as "cultural identity". Is this the same as "cultural difference"?

5. Institutional racism and systemic racism seem to be used interchangeably. There is literature to suggest that institutional racism is a component of systemic racism. Please clarify and/or use the same terminology consistently. Inconsistent use of "IHE" as an abbreviation.

6. In lines 156-157, what are the conclusions that the studies reach about how racism is discussed in statements?

7. In the results and discussion, state the domain instead of referencing the number in Table 2 to prevent the reader from having to keep referencing the table. Sometimes in the discussion the domain is named and other times it isn't. Figure 1 poses the same issue regarding domain names--they are not included and there isn't a legend for each panel.

8. Prejudice is used for the first time in line 750.

9. Minor suggestions: RFT and CRT are spelled out and abbreviated early in the text and later in the text (line 650) both theories are spelled out again.

10. "Tropes" is introduced into the discussion late in the text (line 806) in reference to IHEs being "invested in tropes" which is not a common use of the term in a racial context.

We look forward to receiving your revised manuscript.

Kind regards,

Hanna Landenmark

Staff Editor, PLOS ONE

on behalf of

Nikki R. Wooten

Academic Editor

PLOS ONE

and 

Maurizio Fiaschetti

Academic Editor

PLOS ONE
---

## [Author Response · Author response to Decision Letter 2]

27 Mar 2023

We would like to thank the editors once again for their detailed comments and suggestions. We believe we have benefitted from a diligent re-working of the paper. Details of the changes we have made are below.

1. There is some inconsistency in the use of terminology: colorblindness vs colorblind racism vs colorblind ideology vs colorblind racist ideology vs colorblind rhetoric. This is one example. Authors should clarify if these terms are being used interchangeably or if these terms have different meaning. A table of definitions could address this issue and include other terms: diversity, diversity orthodoxy, "interest convergence", "voice-of-color" thesis, prejudice, and other race-related terms throughout the text.

Thank you for bringing this to our attention. We have resolved this concern in several ways. First, we clarify on page 1 [Lines 32-33] that the ideology of colorblind racism is also referred more simply as colorblind ideology, colorblindness, or colorblind racism in the literature. We include some citations to support our claim. 

Second, we have ensured that we use only these terms when we refer to the ideology of colorblind racism anywhere in the manuscript. To this end, we have made minor modifications across the manuscript wherever we used terms different from these. For instance, we replaced “colorblind rhetoric” in the abstract with “rhetoric consistent with colorblind racism.” Likewise, we replaced the term, “colorblind tropes” with “themes consistent with colorblind ideology.”

Third, as per your suggestion, we have created a Glossary Table included at the end of the manuscript, where we define the most significant terms related to race and methodology used in the manuscript.

2. On page 1, there is a vague and convoluted explanation of terminology from a macro perspective (i.e., institutions of higher education) whereas there tend to be more clearly defined from micro or interpersonal perspectives (e.g., colorblindness at the individual level). Given the sensitivity of DEI and race-related terms, vague definitions can do more harm than good.

We addressed this comment by providing further context on how colorblind ideology manifests in interpersonal racism. Specifically, we begin the second paragraph [Line 43] of the manuscript by stating, “In interpersonal interactions, colorblind racism manifests through the denial or minimization of the role of race in shaping an individual’s experiences or outcomes, and through assertions that any advantages gained by a social group are obtained through merit rather than privileges associated with racial identity [1-2, 5-6].”

3. Lines 46-48, seem to reference the use of microaggressions by college students today, but does not directly use the term "microaggression" which is well-defined in the literature and used indirectly to make racially bias insults in professional and institutional settings.

While we are sympathetic to the editor’s comment, we re-read all the papers cited in that paragraph again and could not find the term microaggressions used anywhere in the cited literature. We believe this is because those papers are principally concerned with attitudes rather than actions, which is where the term microaggressions tends to be central. Accordingly, we do not add a discussion on microaggressions because our paper is similarly primarily focused on institutional positionality and posturing rather than action. 

4. Lines 67-68, which states that "...race is viewed as a matter of cultural difference..." needs further explanation and clarification. This should not be difficult given that 5 articles are cited. Perhaps a definition of diversity should begin this discussion. In line 726, race is referred to as "cultural identity". Is this the same as "cultural difference"?

We have amended the paragraph beginning on Line 72 by adding a definition of diversity as typically used by organizations. Specifically, we state:

“In contrast to colorblindness, diversity, the second dominant theme in race discourse in the United States today, underscores the significance of race rather than minimizing it. Generally speaking, ‘diversity’ has been used by organizations, including IHEs, to refer to heterogeneity of persons based on a myriad of social and personal differences such as race, gender, ethnicity, nationality, and disability status [19-23]. In this context, akin to ethnicity, race is framed as a valued ‘cultural identity’ and racial differences, much like ethnic ones, are viewed as a matter of cultural heterogeneity associated with variability in behaviors, expressions, beliefs, and practices. Racial diversity is thus seen as creating conditions for heterogeneity of interactions among community members, which, in turn, are framed as generative of instrumental benefits such as a superior social climate and creativity of thought [24].” 

As we have clarified that organizational usage of diversity treats race as a cultural identity and made the connection between cultural identity and cultural difference, we have retained the description of race as a ‘cultural identity’ later in the manuscript.

5. Institutional racism and systemic racism seem to be used interchangeably. There is literature to suggest that institutional racism is a component of systemic racism. Please clarify and/or use the same terminology consistently.

We have removed all mentions of ‘institutional racism’ from the manuscript.

6. Inconsistent use of "IHE" as an abbreviation.

We have made numerous changes throughout the manuscript to use ‘IHE’ as consistently as possible. We have retained other terms when they were more accurate or when they fit the prose better.

7. In lines 156-157, what are the conclusions that the studies reach about how racism is discussed in statements?

We have clarified that the conclusions reached by these studies are discussed next by combining the paragraph that came after with the current paragraph [Lines 173-180]. We hope this makes things clearer for the reader. Specifically, we say:

“Regardless of sample, these studies come to comparable conclusions about the ways in which racism is discussed in statements. Specifically, consistent with diversity orthodoxy, researchers find the themes of ‘justice,’ ‘diversity,’ and ‘inclusion’ to be featured prominently in IHEs rhetoric. In their analysis of statements released by 56 leading United States medical schools, Kiang and Tsai [14], for example, find that 40 use the term “inclusion,” 33 use “diversity,” and 29 use “justice.” The authors also note that all institutions used some form of what they characterize as ‘hopeful’ language – rhetoric that invokes diversity as having positive instrumental value.”

8. In the results and discussion, state the domain instead of referencing the number in Table 2 to prevent the reader from having to keep referencing the table. Sometimes in the discussion the domain is named and other times it isn't. Figure 1 poses the same issue regarding domain names--they are not included and there isn't a legend for each panel.

Thank you for bringing this to our attention. We have made several changes to the Analysis and Discussion sections by replacing domain and topic numbers with their substantive names or by adding domain and topic names in parentheses. 

We have offered greater clarity regarding the legends for Figure 1 in the caption for the figure. We have also added a row at the bottom of the figure with Domain names. We tried to fit the domain names into the figures, but that was not possible due to the length of the names.

9. Prejudice is used for the first time in line 750.

We now use the term prejudice several times starting with the first paragraph. However, for clarity, we have added the term to the glossary.

10. Minor suggestions: RFT and CRT are spelled out and abbreviated early in the text and later in the text (line 650) both theories are spelled out again.

We have used the acronyms consistently wherever we refer explicitly to RFT and CRT after their first usage in the manuscript. We have retained the longer forms on occasion when it flowed better with the prose or to remind readers of the meaning of the abbreviation.

11. "Tropes" is introduced into the discussion late in the text (line 806) in reference to IHEs being "invested in tropes" which is not a common use of the term in a racial context.

We have replaced the term ‘tropes’ with ‘themes.’

---

## [Editor Report · Decision Letter 3]

28 Jun 2023

PONE-D-22-17413R3

From Colorblind to Systemic Racism: Emergence of a Rhetorical Shift in Higher Education Discourse in Response to the Murder of George Floyd

PLOS ONE

Dear Dr. Gondal,

Thank you for submitting your manuscript to PLOS ONE. After careful consideration, we feel that it has merit but does not fully meet PLOS ONE’s publication criteria as it currently stands. 

Thank you very much for the revisions you have conducted so far. We have one final comment we would like to raise:

1. Lines 46-48, infer use of microaggressions by college students today, but does not specifically use the term "microaggression" which is well-defined in the literature and used to identify to racially bias insults in professional and institutional settings. The intent of the comment was not for the authors to include a discussion on microagressions, but to explicitly state the behavior to which they are referring in the text whether they are referring to microaggressions or other behavior. The original comment indicated that the authors did not use the terminology "microagression". 

We look forward to receiving your revised manuscript.

Kind regards,

Hanna Landenmark

Staff Editor, PLOS ONE

on behalf of

Nikki R. Wooten and Maurizio Fiaschetti

Academic Editor

PLOS ONE
---

## [Author Response · Author response to Decision Letter 3]

29 Jun 2023

“Thank you very much for the revisions you have conducted so far. We have one final comment we would like to raise:

1. Lines 46-48, infer use of microaggressions by college students today, but does not specifically use the term "microaggression" which is well-defined in the literature and used to identify to racially bias insults in professional and institutional settings. The intent of the comment was not for the authors to include a discussion on microagressions, but to explicitly state the behavior to which they are referring in the text whether they are referring to microaggressions or other behavior. The original comment indicated that the authors did not use the terminology "microagression".”

We have addressed this comment by adding material linked directly to microaggressions. We draw on a seminal paper by Sue et al. (2007) and a more recent paper by Skinner-Dorkenoo et al. (2021) to link behavior associated with color blind racism with microinvalidations, a form of microaggression. Specifically, on pp. 1-2, lines 43-50, we now say:

“In interpersonal interactions, colorblind racism manifests through the denial or minimization of the role of race in shaping an individual’s experiences or outcomes, what Sue et al. [7] describe as microinvalidation, a form of microaggression. Microinvalidations involve denying the feelings, perceptions, observations, or realities of people of color, processes that contribute to reproducing colorblind racism by suppressing the effects of racism and making it more difficult to identify [8]. This typically occurs through assertions that any advantages or disadvantages sustained by a social group are obtained through merit or its lack, rather than privileges or disprivileges associated with racial identity [1-2, 5-6].”

We have also added a comprehensive definition of microaggressions to the glossary table, which appears at the end of the manuscript.

---

## [Editor Report · Decision Letter 4]

21 Jul 2023

From Colorblind to Systemic Racism: Emergence of a Rhetorical Shift in Higher Education Discourse in Response to the Murder of George Floyd

PONE-D-22-17413R4

Dear Dr. Gondal,

We’re pleased to inform you that your manuscript has been judged scientifically suitable for publication and will be formally accepted for publication once it meets all outstanding technical requirements.

Kind regards,

Nikki R. Wooten, PhD

Academic Editor

PLOS ONE
---

## [Editor Report · Acceptance letter]

25 Jul 2023

PONE-D-22-17413R4 

From colorblind to systemic racism: Emergence of a rhetorical shift in higher education discourse in response to the murder of George Floyd 

Dear Dr. Gondal:

I'm pleased to inform you that your manuscript has been deemed suitable for publication in PLOS ONE. Congratulations! Your manuscript is now with our production department. 

Kind regards, 

on behalf of

Dr. Nikki R. Wooten 

Academic Editor

PLOS ONE